# Counterfactual Effect Decomposition in Multi-Agent Sequential Decision Making

Stelios Triantafyllou [1]   Aleksa Sukovic [1]   Yasaman Zolfimoselo [1]   Goran Radanovic [1]

## Abstract

We address the challenge of explaining counterfactual outcomes in multi-agent Markov decision processes. In particular, we aim to explain the total counterfactual effect of an agent's action on the outcome of a realized scenario through its influence on the environment dynamics and the agents' behavior. To achieve this, we introduce a novel *causal explanation formula* that decomposes the counterfactual effect by attributing to each agent and state variable a score reflecting their respective contributions to the effect. First, we show that the total counterfactual effect of an agent's action can be decomposed into two components: one measuring the effect that propagates through all subsequent agents' actions and another related to the effect that propagates through the state transitions. Building on recent advancements in causal contribution analysis, we further decompose these two effects as follows. For the former, we consider *agent-specific effects* – a causal concept that quantifies the counterfactual effect of an agent's action that propagates through a subset of agents. Based on this notion, we use Shapley value to attribute the effect to individual agents. For the latter, we consider the concept of *structure-preserving interventions* and attribute the effect to state variables based on their "intrinsic" contributions. Through extensive experimentation, we demonstrate the interpretability of our approach in a Gridworld environment with LLM-assisted agents and a sepsis management simulator.

## 1. Introduction

Applying counterfactual reasoning to retrospectively analyze the impact of different actions in decision making scenarios is fundamental for accountability. For instance, counterfactual reasoning can be employed to identify *actual causes* (Halpern, 2016; Triantafyllou et al., 2022), attribute *responsibility* (Chockler & Halpern, 2004; Friedenberg & Halpern, 2019), generate *explanations* (Madumal et al., 2020), evaluate *fairness* (Kusner et al., 2017; Huang et al., 2022) and measure *harm* (Richens et al., 2022; Beckers et al., 2022). To achieve such objectives, many studies often rely on the notion of *total counterfactual effects*, which quantifies the extent to which an alternative action would have affected the outcome of a realized scenario.

In multi-agent sequential decision making, an agent's action typically affects the outcome indirectly. To illustrate this, consider the problem of AI-assisted decision making in healthcare (Lynn, 2019), where a clinician and their AI assistant treat a patient over a period of time. Fig. 1a depicts a specific example, where treatment fails. We estimate that if the clinician had not followed the AI's recommendation at step 10 and administered vasopressors (V) instead of mechanical ventilation (E), the treatment would have been successful with an 82% likelihood. Therefore, the considered alternative action admits a high total counterfactual effect. This effect, however, propagates through all subsequent actions of the clinician and the AI, as well as all the changes in the patient's state. This makes the interpretability of the effect more nuanced, as the change from action to outcome can be transmitted by multiple distinct causal mechanisms. In this work, we ask:

*How to explain an action's total counterfactual effect in multi-agent sequential decision making?*

Much prior work in causality has focused on decomposing causal effects (Pearl, 2001; Zhang & Bareinboim, 2018a;b) under the rubric of *mediation analysis* (Imai et al., 2010; 2011; Hicks & Tingley, 2011; VanderWeele, 2016), which aims to understand how effects propagate through causal paths. However, such an approach would not yield interpretability in multi-agent sequential decision making. There can be exponentially many paths connecting an action to the outcome, and not all of them have a clear operational meaning to help explain the effect intuitively. We instead posit that it is more natural to interpret the effect of an action in terms of its influence on the agents' behavior and the

---

[1]Max Planck Institute for Software Systems, Germany. Correspondence to: Stelios Triantafyllou <strianta@mpi-sws.org>.

*Proceedings of the 42nd International Conference on Machine Learning*, Vancouver, Canada. PMLR 267, 2025. Copyright 2025 by the author(s).

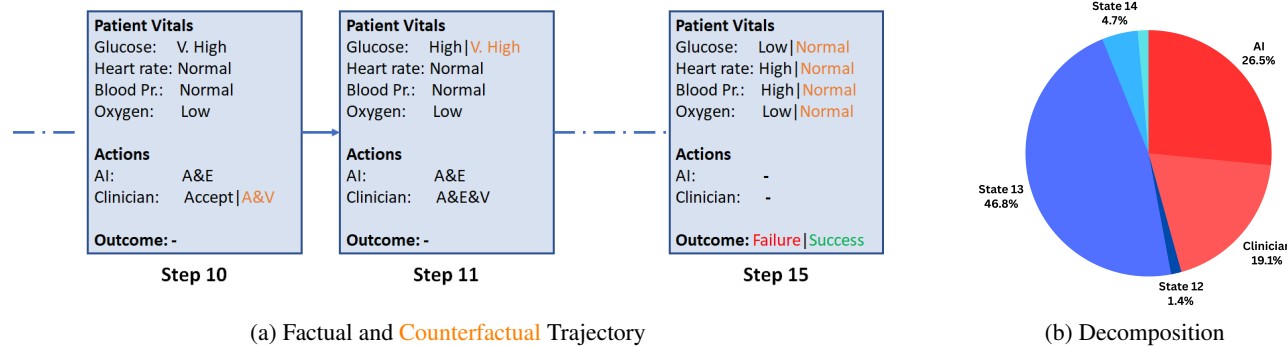

(a) Factual and Counterfactual Trajectory

(b) Decomposition

Figure 1: Fig. 1a depicts (part of) a simulated scenario from the two-agent *Sepsis* environment in Section 6.2, where the patient's treatment fails. In the same figure, we have also included the values from a sampled counterfactual scenario (values that are different are shown in orange), where the clinician's action is fixed to override the AI's action at step 10. Hence, the patient receives treatment A&V instead of A&E. Plot 1b shows the results of our decomposition approach for this scenario.

environment dynamics. Therefore, we need to analyze how the effect propagates through: (a) the subsequent agents' actions and (b) the state transitions of the environment. In the previous example, the total counterfactual effect of the considered action can be decomposed as shown in Plot 1b. This approach explains the effect by attributing a score to each doctor (clinician and AI) and patient state, reflecting their respective contributions to the overall effect.

**Contributions.** Focusing on Multi-Agent Markov Decision Processes and Structural Causal Models, we provide a systematic approach to attributing the total counterfactual effect of an agent's action on the outcome of a given trajectory, based on the following bi-level decomposition.

(**Level 1**) We first introduce a *causal explanation formula*, which decomposes the total counterfactual effect of an agent's action into the *total agent-specific effect* and the *reverse state-specific effect*. The former refers to the effect that propagates through all subsequent agents' actions, and is formulated via the recently introduced notion of *agent-specific effects* (Triantafyllou et al., 2024). The latter refers to the effect that would have been lost or gained had the action not been propagated through the state transitions, and it is a special case of *path-specific effects* (Avin et al., 2005).

(**Level 2a**) To further decompose the total agent-specific effect (tot-ASE), we propose an axiomatic framework based on agent-specific effects for attributing the total effect to individual agents. The set of axioms includes *efficiency*, which requires that agents' contributions sum up to tot-ASE. We show how to operationalize Shapley value with agent-specific effects, to obtain a method for decomposing tot-ASE, which uniquely satisfies the set of proposed axioms.

(**Level 2b**) To further decompose the reverse state-specific effect (r-SSE), we utilize the notion of *intrinsic causal contributions* (ICC) (Janzing et al., 2024). ICC enables us to quantify the informativeness of individual state variables

regarding the counterfactual outcomes needed for the computation of r-SSE. We propose a method for decomposing r-SSE that is *efficient* under a relatively mild assumption that at least one state variable has non-zero ICC (i.e., is informative about the counterfactual outcomes).

We experimentally validate the interpretability of our approach using two multi-agent environments: a grid-world environment, where two RL actors are instructed by an LLM planner to complete a sequence of tasks, and the sepsis management simulator from Fig. 1.[1]

### 1.1. Additional Related Work

This paper is related to works on mediation analysis and especially to those that consider multiple (sequential) mediators (Daniel et al., 2015; Steen et al., 2017; VanderWeele & Vansteelandt, 2014; Chiappa, 2019). As mentioned earlier, the main distinction between this line of work and ours is that we analyze how effects propagate through agents and state variables in an MMDP, instead of causal paths in general SCMs. In a similar sense, our work also relates to the areas of *causal contributions* (Janzing et al., 2024; Jung et al., 2022; Heskes et al., 2020) and *flow-based* attribution methods (Singal et al., 2021; Wang et al., 2021). The former studies how to attribute a target effect to different causes (often model features) based on their degree of some notion of contribution to that effect. The latter considers the problem of assigning credit to the edges of a causal graph, instead of the nodes, for explaining causal effects.

## 2. Background and Formal Framework

In this section, we present our formal framework, which is adopted from (Triantafyllou et al., 2024) and builds on Multi-

---

[1]Code to reproduce our experiments is available at https://github.com/stelios30/cf-effect-decomposition.git.

Agent Markov Decision Processes (MMDPs) (Boutilier, 1996) and Structural Causal Models (SCMs) (Pearl, 2009). A table summarizing the notation is provided in Appendix B. Appendix N provides a graphical illustration of all counterfactual effects discussed in this section and the next.

## 2.1. Multi-Agent Markov Decision Processes

An MMDP is represented as a tuple $\langle \mathcal{S}, \{1, ..., n\}, \mathcal{A}, T, h, \sigma \rangle$, where: $\mathcal{S}$ is the state space; $\{1, ..., n\}$ is the set of agents; $\mathcal{A} = \times_{i=1}^{n} \mathcal{A}_i$ is the joint action space, with $\mathcal{A}_i$ being the action space of agent $i$; $T : \mathcal{S} \times \mathcal{A} \times \mathcal{S} \rightarrow [0, 1]$ is the transition probability function; $h$ is the finite time horizon; $\sigma$ is the initial state distribution.[2] Each agent $i \in \{1, ..., n\}$ has a stationary decision-making policy $\pi_i$, with the joint policy of all agents represented as $\pi$. The probability of agents jointly taking action $\mathbf{a}_t = (a_{1,t}, ..., a_{n,t})$ in state $s_t$ at time $t$ is thus given by $\pi(\mathbf{a}_t|s_t) = \pi_1(a_{1,t}|s_t) \cdots \pi_n(a_{n,t}|s_t)$, while the probability of transitioning from state $s_t$ to state $s_{t+1}$ is determined by $T(s_{t+1}|s_t, \mathbf{a}_t)$. A sequence of such state-action pairs $\{(s_t, \mathbf{a}_t)\}_{t \in \{0,...,h-1\}}$ and final state $s_h$ is called a trajectory. With $\tau(X)$, we denote the value of variable $X$ in trajectory $\tau$.

## 2.2. MMDPs and Structural Causal Models

We utilize the MMDP-SCM framework (Triantafyllou et al., 2024) to express an MMDP coupled with a joint policy $\pi$ as an SCM. Specifically, an MMDP-SCM $\langle \mathbf{V}, \mathbf{U}, P(\mathbf{u}), \mathcal{F} \rangle$ consists of:

(i) a tuple $\mathbf{V} = \langle S_0, A_{1,0}, ..., A_{n,0}, ..., S_h \rangle$ of the observed variables whose causal relations are modeled, i.e., all state and action variables of the MMDP;

(ii) a tuple $\mathbf{U} = \langle U^{S_0}, U^{A_{1,0}}, ..., U^{A_{n,0}}, ..., U^{S_h} \rangle$ of mutually independent unobserved noise variables which capture any underlying stochasticity of the MMDP and agents' policies;

(iii) A joint probability distribution $P(\mathbf{u}) = \prod_{u^i \in \mathbf{u}} P(u^i)$ over $\mathbf{U}$;

(iv) A collection $\mathcal{F}$ of deterministic functions that determine the values of all observed variables in $\mathbf{V}$ via the following *structural equations*

$$S_0 := f^{S_0}(U^{S_0}); \quad S_t := f^S(S_{t-1}, \mathbf{A}_{t-1}, U^{S_t});$$
$$A_{i,t} := f^{A_i}(S^t, U^{A_{i,t}}). \tag{1}$$

Note that any context $\mathbf{u} \sim P(\mathbf{u})$ induces a unique trajectory $\tau$, such that $\forall X \in \mathbf{V}$ it holds that $\tau(X)$ is the solution of $X$, for the particular $\mathbf{u}$, in the MMDP-SCM. Furthermore,

similar to general SCMs, the MMDP-SCM induces a directed *causal graph*, which can be found in Appendix D. Appendix C also describes the conditions under which the observational distribution of an MMDP-SCM is consistent with some MMDP and joint policy. In this paper, we focus on categorical MMDP-SCMs.

## 2.3. Interventions and Counterfactuals

Consider an MMDP-SCM $M$. An *intervention* on the action variable $A_{i,t}$ of $M$ corresponds to the process of modifying the structural equation $A_{i,t} := f^{A_i}(S^t, U^{A_{i,t}})$ from Eq. 1. More specifically, a **hard intervention** $do(A_{i,t} := a_{i,t})$ fixes the value of $A_{i,t}$ to the constant $a_{i,t}$, resulting in a new MMDP-SCM denoted by $M^{do(A_{i,t}:=a_{i,t})}$. Similar to (Correa et al., 2021), when random variables have subscripts we will use square brackets to denote interventions.

Let now $Z \in \mathbf{V}$ and $\mathbf{u} \sim P(\mathbf{u})$. We denote with $Z_{do(A_{i,t}:=a_{i,t})}(\mathbf{u})$ or $Z_{a_{i,t}}(\mathbf{u})$ for short, the solution of $Z$ for $\mathbf{u}$ in $M^{do(A_{i,t}:=a_{i,t})}$, and with $Z_{a_{i,t}}$ the random variable induced by averaging over $\mathbf{U}$. Typically, $Z_{a_{i,t}}(\mathbf{u})$ is referred to as the *potential response* of $Z$ to $do(A_{i,t} := a_{i,t})$. A **natural intervention** $do(A_{j,t'} := A_{j,t'[a_{i,t}]})$ replaces the structural equation of $A_{j,t'}$ in $M$ with the potential response of $A_{j,t'}$ to the (hard) intervention $do(A_{i,t} := a_{i,t})$.

Given a trajectory $\tau$ and a *response* variable $Y \in \mathbf{V}$, the **counterfactual probability** $P(Y_{a_{i,t}} = y|\tau)_M$ or $P(y_{a_{i,t}}|\tau)_M$ for short, measures the probability of $Y$ taking the value $y$ in $\tau$ had $A_{i,t}$ been set to $a_{i,t}$. Subscript $M$ here implies that the probability is defined over the MMDP-SCM $M$. When necessary, $P(\cdot|\tau; M)$ is used to denote that $\tau$ was generated by $M$. Next, we define a standard causal notion that is used to quantify the counterfactual impact of intervention $do(A_{i,t} := a_{i,t})$ on $Y$.[3]

**Definition 2.1** (TCFE). Given an MMDP-SCM $M$ and a trajectory $\tau$ of $M$, the *total counterfactual effect* of intervention $do(A_{i,t} := a_{i,t})$ on $Y \in \mathbf{V}$, relative to reference $\tau(A_{i,t})$, is defined as

$$\text{TCFE}_{a_{i,t}, \tau(A_{i,t})}(Y|\tau)_M = \mathbb{E}[Y_{a_{i,t}}|\tau]_M - \mathbb{E}[Y_{\tau(A_{i,t})}|\tau]_M$$
$$= \mathbb{E}[Y_{a_{i,t}}|\tau]_M - \tau(Y).$$

**Assumptions and counterfactual identifiability.** Note that there might be multiple MMDP-SCMs whose observational distribution is consistent with some MMDP-joint policy pair, but yield different counterfactuals, e.g., different values for TCFE. This means that without further assumptions, counterfactuals cannot be identified from observations alone. To enable *counterfactual identifiability*, we thus make the

---

[2]For ease of notation, rewards are considered part of the states.

[3]In this paper, we consider counterfactual effects mostly relative to the factual action $\tau(A_{i,t})$. However, we note that generally any valid action can be used as a reference value.

following assumptions. First, we consider unobserved variables to be mutually independent. Second, we assume that MMDP-SCMs satisfy the (weak) *noise monotonicity* condition introduced by (Triantafyllou et al., 2024). Note that the latter assumption is not limiting for the MMDP distribution or the agents' policies, i.e., every MMDP can be consistently represented by a noise-monotonic MMDP-SCM. What is restricted instead is the expressivity of the model's counterfactual distribution. Details about noise monotonicity can be found in Appendix E.

## 3. Decomposing Total Counterfactual Effect

The total counterfactual effect can inform us about the extent to which an alternative action would have affected the outcome of a trajectory. However, this measure alone does not provide any further insights on *why* or *how* that action would have affected the outcome. In this section, we introduce a novel causal explanation formula that decomposes TCFE w.r.t. the two building blocks of an MMDP – its states and its agents. First, based on prior work we define two causal quantities for measuring how much of the total counterfactual effect of an agent's action on some response variable is mediated by (a) all future agents' actions and (b) the subsequent MMDP's state transitions.

**Definition 3.1** (tot-ASE)**.** Given an MMDP-SCM $M$ and a trajectory $\tau$ of $M$, the *total agent-specific effect* of intervention $do(A_{i,t} := a_{i,t})$ on $Y \in \mathbf{V}$, relative to reference $\tau(A_{i,t})$, is defined as

$$
\begin{aligned}
\text{ASE}_{a_{i,t},\tau(A_{i,t})}^{\{1,\dots,n\}}(Y|\tau)_M &= \\
&= \mathbb{E}[Y|\tau; M]_{M^{do(I)}} - \mathbb{E}[Y_{\tau(A_{i,t})}|\tau]_M \\
&= \mathbb{E}[Y|\tau; M]_{M^{do(I)}} - \tau(Y),
\end{aligned}
$$

where $I = \{A_{i',t'} := A_{i',t'[a_{i,t}]}\}_{i' \in \{1,\dots,n\}, t'>t}$.

**Definition 3.2** (SSE)**.** Given an MMDP-SCM $M$ and a trajectory $\tau$ of $M$, the *state-specific effect* of intervention $do(A_{i,t} := a_{i,t})$ on $Y \in \mathbf{V}$, relative to reference $\tau(A_{i,t})$, is defined as

$$
\begin{aligned}
\text{SSE}_{a_{i,t},\tau(A_{i,t})}(Y|\tau)_M &= \\
&= \mathbb{E}[Y_{a_{i,t}}|\tau; M]_{M^{do(I)}} - \mathbb{E}[Y_{\tau(A_{i,t})}|\tau]_M \\
&= \mathbb{E}[Y_{a_{i,t}}|\tau; M]_{M^{do(I)}} - \tau(Y),
\end{aligned}
$$

where $I = \{A_{i',t'} := A_{i',t'[\tau(A_{i',t'})]}\}_{i' \in \{1,\dots,n\}, t'>t}$.

In words, Definition 3.1 measures the difference between the factual value of $Y$, i.e., $\tau(Y)$, and the (expected) counterfactual value of $Y$ had all agents taken the actions that they would naturally take under intervention $do(A_{i,t} := a_{i,t})$ after time-step $t$. On the other hand, Definition 3.2 measures the counterfactual effect of intervention $do(A_{i,t} := a_{i,t})$ on $Y$ in a modified model where all subsequent agents' actions

are fixed to their factual values, i.e., their actions in $\tau$. Definition 3.1 is in line with the notion of *agent-specific effects* introduced by (Triantafyllou et al., 2024) and revisited here in Section 5, while SSE can be seen as a special case of *path-specific effects* (Avin et al., 2005).

Perhaps counter to intuition, it does not hold that TCFE is always decomposed into tot-ASE and SSE. An empirical counter-example for this is provided in Section 6.1. However, by comparing Definitions 3.1 and 3.2, we observe that the total agent-specific effect associated with the transition from the factual action $\tau(A_{i,t})$ to the counterfactual action $a_{i,t}$ is closely related to the state-specific effect associated with the reverse transition, i.e., the effect

$$
\begin{aligned}
\text{SSE}_{\tau(A_{i,t}),a_{i,t}}(Y|\tau)_M &= \\
&= \mathbb{E}[Y_{\tau(A_{i,t})}|\tau; M]_{M^{do(I)}} - \mathbb{E}[Y_{a_{i,t}}|\tau]_M \\
&= \mathbb{E}[Y|\tau; M]_{M^{do(I)}} - \mathbb{E}[Y_{a_{i,t}}|\tau]_M, \quad (2)
\end{aligned}
$$

where $I = \{A_{i',t'} := A_{i',t'[a_{i,t}]}\}_{i' \in \{1,\dots,n\}, t'>t}$. We will refer to the latter as the *reverse state-specific effect* or r-SSE for short, to clearly distinguish it from SSE. In words, r-SSE measures the difference in the counterfactual value of $Y$ under intervention $do(A_{i,t} := a_{i,t})$, assuming that the state $S_{t+1}$ had not been affected by the intervention, but all subsequent agents' actions had. Based on this observation, we derive the following decomposition of the total counterfactual effect.

**Theorem 3.3.** *The total counterfactual effect, total agent-specific effect and reverse state-specific effect obey the following relationship*

$$
\begin{aligned}
\text{TCFE}_{a_{i,t},\tau(A_{i,t})}(Y|\tau) &= \\
&= \text{ASE}_{a_{i,t},\tau(A_{i,t})}^{\{1,\dots,n\}}(Y|\tau) - \text{SSE}_{\tau(A_{i,t}),a_{i,t}}(Y|\tau). \quad (3)
\end{aligned}
$$

Theorem 3.3 states that the total counterfactual effect of $do(A_{i,t} := a_{i,t})$ on $Y$ equals to the effect that propagates only through the agents *minus* the effect that would have been lost or gained had the intervention not been propagated through the states.

**Connection to prior work.** Our result is similar in principle with the well-known *causal mediation formula* for arbitrary SCMs (Pearl, 2001), which decomposes the *total causal effect* of an intervention into the *natural direct* and *indirect effects*. Thus, Theorem 3.3 can be viewed as an extension of Theorem 3 from (Pearl, 2001), applied to the problem of counterfactual effect decomposition in multi-agent MDPs.

**Sepsis example.** Going back to our example scenario from the introduction, the result of our decomposition can be interpreted as follows: (a) 45.6% of the TCFE is attributed to how the AI and the clinician would have responded to the intervention (tot-ASE $\approx 0.374$); (b) the remaining 54.4% is attributed to the influence that the intervention has on the patient state ($-$r-SSE $\approx 0.446$).

# 4. Decomposing Reverse State-Specific Effect

In this section, we focus on further decomposing the reverse state-specific effect. More specifically, our goal is to attribute to each state variable a score reflecting its contribution to r-SSE. Our approach utilizes the notion of *intrinsic causal contributions* (ICC) introduced by (Janzing et al., 2024). For general SCMs, the ICC of an observed variable $X$ to a target variable $Y$ measures the reduction of uncertainty in $Y$ when conditioning on the noise variable $U^X$. In our work, we model uncertainty using the expected conditional variance and modify the ICC definition to quantify the influence of state variables on the variation of r-SSE.

Let $k \in \{0, ..., h\}$. We denote with $\mathbf{U}^{S_k}$ the set $\langle U^{S_k}, U^{A_{1,k}}, ..., U^{A_{n,k}} \rangle$ for $k < h$, and with $\mathbf{U}^{S_h}$ the set $\langle U^{S_h} \rangle$. We also denote with $\mathbf{U}^{<S_k}$ the set of noise terms associated with the observed variables preceding (chronologically) $S_k$. Note that r-SSE essentially measures the expected value of the difference $\Delta Y_{I,a_{i,t}} = Y_I - Y_{a_{i,t}}$ in $M$, when noise terms $\mathbf{U}$ are sampled from the posterior distribution $P(\mathbf{u}|\tau)$. Thus, the ICC can be defined in our context as follows

$$\text{ICC}(S_k \to \Delta Y_{I,a_{i,t}}|\tau) = \text{Unc}^{<S_k} - \text{Unc}^{\leq S_k}, \quad (4)$$

$$\text{where } \text{Unc}^{<S_k} = \mathbb{E}[\text{Var}(\Delta Y_{I,a_{i,t}}|\tau, \mathbf{U}^{<S_k})|\tau],$$
$$\text{Unc}^{\leq S_k} = \mathbb{E}[\text{Var}(\Delta Y_{I,a_{i,t}}|\tau, \mathbf{U}^{<S_k}, \mathbf{U}^{S_k})|\tau],$$

and $I = \{A_{i',t'} := A_{i',t'[a_{i,t}]}\}_{i' \in \{1,...,n\}, t'>t}$. In words, Eq. 4 measures the reduction of variance in $\Delta Y_{I,a_{i,t}}$ caused by conditioning on the noise variables associated with state $S_k$ and the agents' actions taken therein. Based on this, we can now define our attribution method for r-SSE.

**Definition 4.1** (r-SSE-ICC). Given an MMDP-SCM $M$ and a trajectory $\tau$ of $M$, r-SSE-ICC assigns to each state variable $S_k$ for $k \in \{0, ..., h\}$ a contribution score for the reverse state-specific effect $\text{SSE}_{\tau(A_{i,t}),a_{i,t}}(Y|\tau)_M$, equal to

$$\psi_{S_k} := \frac{\text{ICC}(S_k \to \Delta Y_{I,a_{i,t}}|\tau)_M}{\text{Var}(\Delta Y_{I,a_{i,t}}|\tau)} \cdot \text{SSE}_{\tau(A_{i,t}),a_{i,t}}(Y|\tau)_M,$$

if the unconditional variance $\text{Var}(\Delta Y_{I,a_{i,t}}|\tau) > 0$, and equal to 0 otherwise.

According to Definition 4.1, the reverse state-specific effect is allocated among state variables in proportion to their intrinsic contribution to the effect. Intuitively, this means that the influence of a state variable to the r-SSE is represented by the relative degree to which we can more precisely estimate the effect if we could also predict the counterfactual value of that state under the interventions $do(I)$ and $do(A_{i,t} := a_{i,t})$. Thus, if knowing what would have happened in state $S_k$ is pivotal for the accuracy of our counterfactual prediction then contribution score $\psi_{S_k}$ would be high, whereas if it has small influence then $\psi_{S_k}$ would be closer to zero. In the

case where we can exactly compute $\text{SSE}_{\tau(A_{i,t}),a_{i,t}}(Y|\tau)_M$ without conditioning on any noise term, e.g., if environment and policies are deterministic, then our approach does not decompose the r-SSE any further.

**Algorithm.** Appendix G includes an algorithm for the approximation of the expected conditional variance of $\Delta Y_{I,a_{i,t}}$. Our algorithm follows the standard *abduction-action-prediction* methodology for counterfactual inference (Pearl, 2009): it samples conditioning and non-conditioning noise variables independently from the posterior distribution, estimates the noise-conditional variance for r-SSE and returns the average value.

**Causal interpretation.** The attribution method described in Definition 4.1 relies on *do* interventions that are performed only on agents' actions, meaning that the causal mechanisms of the environment remain intact. Conditioning on noise terms can be considered as a form of *structure-preserving* interventions, i.e., interventions that depend on the values of the parents of the exposure variable (here previous state and actions), and do not perturb the observed distribution. For a more detailed discussion on the causal meaning of ICC we refer the reader to Section 3.1 in (Janzing et al., 2024).

**Plain ICC.** Since there is a unique causal order among the state variables of an MMDP-SCM, the time order, there is no arbitrariness due to order-dependence. Thus, we consider the "plain" ICC for our approach instead of the Shapley based symmeterization used in (Janzing et al., 2024).

Finally, we show that r-SSE-ICC fully allocates r-SSE among the states following the intervention.

**Theorem 4.2.** *Let $t_Y$ denote the time-step of response variable $Y$ and $\psi$ be the output of* r-SSE-ICC *for the reverse state-specific effect* $\text{SSE}_{\tau(A_{i,t}),a_{i,t}}(Y|\tau)_M$. *If* $\text{Var}(\Delta Y_{I,a_{i,t}}|\tau) > 0$, *then it holds that* $\sum_{k \in [t+1,t_Y]} \psi_{S_k} = \text{SSE}_{\tau(A_{i,t}),a_{i,t}}(Y|\tau)_M$.

**Sepsis example.** The result of the r-SSE-ICC method in the Sepsis example can be interpreted as follows: if we knew the counterfactual state of the patient at step 13, following the intervention at step 10, we could estimate the reverse state-specific effect with almost no uncertainty. While, knowing the exact counterfactual value for any of the previous states would not lead to a comparable reduction in uncertainty.

# 5. Decomposing Total Agent-Specific Effect

In this section, we focus on further decomposing the total agent-specific effect. More specifically, our goal is to attribute to each agent a score reflecting its contribution to tot-ASE. Our approach is based on a well-established solution concept in cooperative game theory, the *Shapley value* (Shapley, 1953), and it utilizes the notion of *agent-specific effects* introduced by (Triantafyllou et al., 2024).

**Definition 5.1** (ASE). Given an MMDP-SCM $M$, a non-empty subset of agents $\mathbf{N}$ in $M$ and a trajectory $\tau$ of $M$, the $\mathbf{N}$-*specific effect* of intervention $do(A_{i,t} := a_{i,t})$ on $Y \in \mathbf{V}$, relative to reference $\tau(A_{i,t})$, is defined as

$$
\begin{aligned}
\text{ASE}^{\mathbf{N}}_{a_{i,t},\tau(A_{i,t})}(Y|\tau)_M & = \\
& = \mathbb{E}[Y|\tau; M]_{M^{do(I)}} - \mathbb{E}[Y_{\tau(A_{i,t})}|\tau]_M \\
& = \mathbb{E}[Y|\tau; M]_{M^{do(I)}} - \tau(Y),
\end{aligned}
$$

where $I = \{A_{i',t'} := \tau(A_{i',t'})\}_{i'\notin\mathbf{N},t'>t} \cup \{A_{i',t'} := A_{i',t'[a_{i,t}]}\}_{i'\in\mathbf{N},t'>t}$.

In contrast to the total ASE, the $\mathbf{N}$-specific effect quantifies the counterfactual effect of an intervention that propagates only through a subset of agents in the system, the *effect agents*, instead of all agents. Compared to Definition 3.1, here the actions of the *non-effect agents* are set to their factual values. In the context of agent-specific effects studied here, Shapley value can be defined as follows.

**Definition 5.2** (ASE-SV). Given an MMDP-SCM $M$ and a trajectory $\tau$ of $M$, ASE-SV assigns to each agent $j \in \{1,...,n\}$ a contribution score for the total agent-specific effect $\text{ASE}^{\{1,...,n\}}_{a_{i,t},\tau(A_{i,t})}(Y|\tau)_M$, equal to

$$
\begin{aligned}
\phi_j := \sum_{S \subseteq \{1,...,n\}\setminus\{j\}} w_S \cdot \big[ & \text{ASE}^{S\cup\{j\}}_{a_{i,t},\tau(A_{i,t})}(Y|\tau)_M \\
& - \text{ASE}^{S}_{a_{i,t},\tau(A_{i,t})}(Y|\tau)_M \big],
\end{aligned}
$$

where coefficients $w_S$ are set to $w_S = \frac{|S|!(n-|S|-1)!}{n!}$.

Next, we define a number of desirable properties for the attribution of the total agent-specific effect. These properties are inspired from the game theory literature (Jain & Mahdian, 2007; Shoham & Leyton-Brown, 2008; Young, 1985) and translated to our setting.

**Efficiency:** *The total sum of agents' contribution scores is equal to tot-ASE.*

**Invariance:** *Agents who do not contribute to tot-ASE are assigned a zero contribution score.*

**Symmetry:** *Agents who contribute equally to tot-ASE are assigned the same contribution score.*

**Contribution monotonicity:** *The contribution score assigned to an agent depends only on its marginal contributions to tot-ASE and monotonically so.*

We formally state these properties in Appendix F. We now restate in the setting of agent-specific effects studied here an existing uniqueness result for Shapley value.

**Theorem 5.3.** *(Young, 1985)* ASE-SV *is a unique attribution method for the total agent-specific effect that satisfies efficiency, invariance, symmetry and contribution monotonicity.*

**Sepsis example.** ASE-SV attributes tot-ASE to both the AI and the clinician. As illustrated in Plot 1b, a larger portion of the effect is attributed to how the AI would have responded to the intervention.

# 6. Experiments

In this section, we empirically evaluate our approach to counterfactual effect decomposition using two environments, *Gridworld* and *Sepsis*. We refer the reader to Appendix J for more details on our experimental setup and implementation, and to Appendix K for additional results. Throughout both experiments, we use 100 posterior samples for estimating counterfactual effects and 20 additional ones for the conditional variance. Additional experiments evaluating the estimation error of our results as well as their robustness to the noise monotonicity assumption are provided in Appendix L and Appendix M, respectively.

## 6.1. Gridworld with LLM-assisted RL agents

**Environment.** We consider the gridworld depicted in Fig. 2a, where two actors, $\mathcal{A}_1$ and $\mathcal{A}_2$, are tasked with delivering objects. In the beginning of each trajectory, two randomly sampled objects spawn in each of the *boxes* located on the rightmost corners of the gridworld. The color of each object determines its value. Colored cells indicate areas of large stochastic penalty, which is significantly reduced when actors carry an object of matching color. Cells denoted with stars are delivery locations. If an object is delivered to the location with the matching color, then its value is rewarded.

**Implementation.** We adopt a **Planner-Actor-Reporter** system akin to (Dasgupta et al., 2023). Planner is implemented using a pre-trained LLM and few-shot learning, to provide actors with instructions. More specifically, Planner can instruct actors to: *examine* a box, *pickup* an object and *deliver* that object to a specific destination. Furthermore, we assume an optimal Reporter whose task is to report to Planner the necessary information about the state of the environment. In particular, Reporter provides information about the boxes' contents and which objects were picked up by the actors. Finally, the two actors are trained with deep RL to follow the Planner's instructions.

**Setup.** For our demonstration purposes, we consider the (factual) trajectory illustrated in Fig. 2a. We intervene on the pickup action of actor $\mathcal{A}_2$ forcing it to disobey Planner and choose the green object. The resulting counterfactual trajectory can be seen in Fig. 2a as well. Additional results from a second experiment, where we intervene on the Planner's action, can be found in Appendix K.1.

**Counterfactual effects.** To measure the total counterfactual effect in this scenario, we estimate the value of the total

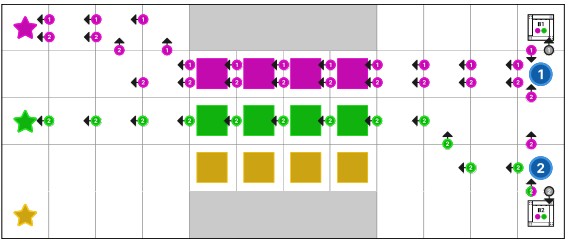

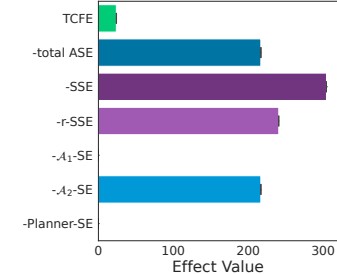

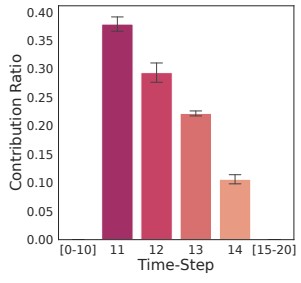

(a) Factual and Counterfactual Trajectory  (b) Counterfactual Effects  (c) r-SSE Attribution

Figure 2: 2a depicts the actors' movements in both the factual and counterfactual trajectory used in our experiments. Initially, both $\mathcal{A}_1$ and $\mathcal{A}_2$ (represented by solid circles) are instructed to pickup the pink object and deliver it to the pink delivery location. In the counterfactual trajectory, $\mathcal{A}_2$ is forced to pickup the green object instead, prompting Planner to issue an alternative instruction for delivery to the green location. This intervention does not affect $\mathcal{A}_1$'s behavior. A textual depiction of both trajectories is provided in Appendix K.1. Plot 2b shows the values of various counterfactual effects computed on the trajectory's **discounted** total reward. The *minus* sign indicates that the negative of these values are plotted. Plot 2c shows the contribution ratios attributed to all state variables by r-SSE-ICC. Averages and standard errors are reported for 5 seeds.

reward collected in the counterfactual trajectory, and subtract from it the observed return. For the total agent-specific effect (Definition 3.1), we need to isolate the effect of the intervention that propagates only through the agents ($\mathcal{A}_1$, $\mathcal{A}_2$ and Planner). Compared to TCFE, we thus estimate the return of the counterfactual trajectory in which the stochastic penalties are realized as if $\mathcal{A}_2$ carries the pink object. For the state-specific effect (Definition 3.2), we have to isolate the effect of the intervention that propagates only through states. Therefore, we estimate the return of the counterfactual trajectory in which agents take their factual actions, but stochastic penalties are realized as if $\mathcal{A}_2$ carries the green object. For the reverse SSE (Eq. 2), it suffices to compute the difference between the returns of the counterfactual trajectories considered for tot-ASE and TCFE.

**Causal explanation formula.** Plot 2b indicates that TCFE is not decomposed into tot-ASE and SSE. Theorem 3.3, on the other hand, is empirically validated in this scenario.

**ASE-SV.** According to Plot 2b, the ASE-SV attributes zero scores to both $\mathcal{A}_1$ and Planner, while assigning the full tot-ASE to $\mathcal{A}_2$. $\mathcal{A}_1$'s lack of contribution to the effect is due to its unresponsiveness to $\mathcal{A}_2$'s actions. Although the Planner does respond to $\mathcal{A}_2$, it is unable to directly influence the environment's state. As a result, the effect of our intervention on the total reward does not propagate through the Planner's actions. These represent two distinct mechanisms by which agents can be excluded from contributing to the total agent-specific effect.

**r-SSE-ICC.** Plot 2c shows that r-SSE-ICC pinpoints four state variables with non-zero contributions to the reverse state-specific effect. As expected, the time-steps of these variables coincide with the time-steps at which $\mathcal{A}_2$ traverses the colored cells in the counterfactual trajectory, as these are

the only sources of stochasticity in the environment. Moreover, we observe that the scores attributed to the four states decrease over time. Since penalties are sampled independently, this can be interpreted as follows: the uncertainty over the counterfactual penalty estimates is greater in earlier time-steps. The latter can be confirmed by comparing the penalty distributions from Table 2 in Appendix J.

### 6.2. Sepsis

**Environment.** The two-agent variant of the sepsis treatment setting (Triantafyllou et al., 2024) we consider here involves a clinician and an AI agent who take sequential actions in a turn-based manner for treating an ICU patient. At each round, the AI recommends one of 8 possible treatments, which is then reviewed and potentially overridden by the clinician. The likelihood of the clinician overriding the AI's treatment at any given state is modeled by a parameter $\mu$, which is varied in our experiment. Intuitively, $\mu$ serves as a proxy for the clinician's level of trust in the AI's recommendations: higher values of $\mu$ correspond to greater levels of trust. If the AI's action is not overridden, then its selected treatment is applied. Otherwise, a new treatment selected by the clinician is applied. The outcome of a trajectory is deemed successful if the patient is kept alive for 20 rounds or gets discharged earlier.

**Evaluation of ASE-SV.** We generate 600 trajectories with unsuccessful outcomes. We then measure the total counterfactual effect of all possible alternative actions on the final state of these trajectories and keep those that exhibit TCFE $\geq 0.8$. Through that process, 8728 alternative actions are selected for the evaluation of ASE-SV. For all selected actions, we compute their total agent-specific effect, *clinician-specific effect* and *AI-specific effect*. As expected,

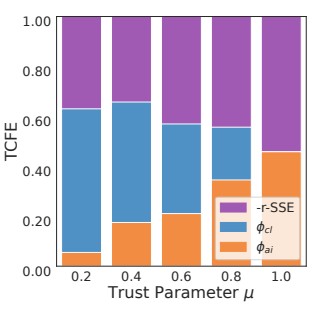

(a) Decomposition: AI

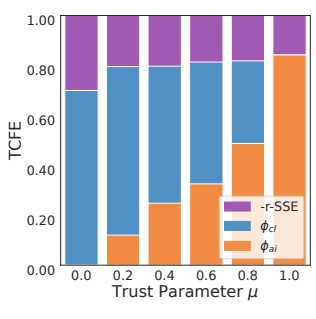

(b) Decomposition: CL

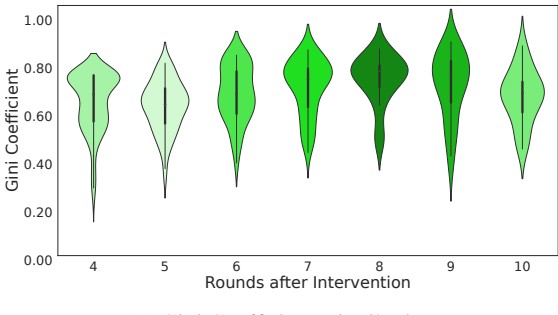

(c) Gini Coefficient Distribution

Figure 3: Plots 3a and 3b show the average percentage decomposition of -r-SSE and scores $\phi_{cl}$ and $\phi_{ai}$ attributed by ASE-SV w.r.t. TCFE, for interventions on the actions of AI and clinician, respectively, while varying trust parameter $\mu$. Plot 3c shows the Gini coefficient distribution over the scores attributed to state variables by the r-SSE-ICC method. The x-axis displays how many rounds after the considered intervention the trajectory terminates.

the sum of the two individual effects does not equal the total one, with discrepancies of up to 95%. In contrast, in our experiments, ASE-SV always attributes the effect *efficiently* to the clinician and AI, as supported by Theorem 5.3.

Plots 3a and 3b show the average percentage composition of the reverse state-specific effect and the agent scores attributed by ASE-SV w.r.t. the total counterfactual effect, for different trust levels. Plot 3a (resp. Plot 3b) considers the average over all selected AI (resp. clinician) actions. Results reveal that our method demonstrates a trend similar to the one described in (Triantafyllou et al., 2024). In particular, the amount of tot-ASE attributed to the clinician (resp. AI) decreases (resp. increases) as the level of trust rises, eventually reaching zero (resp. full) when the clinician completely trusts the AI's recommendations. This observation is intuitive, since the clinician is expected to contribute less to the effect as it acts more infrequently in the environment, while at the same time the AI is expected to contribute more as it assumes greater agency. Thus, we conclude that ASE-SV can efficiently attribute tot-ASE without sacrificing the conceptual power of agent-specific effects.

**Evaluation of r-SSE-ICC.** We consider the same setup as before and categorize all selected actions based on the difference between the round that they were taken and the final round of their respective trajectory. For instance, if the action we consider was taken by the AI at the third round of a trajectory with 8 rounds in total then the *round difference* for that action is 5. For our analysis, we maintain actions with round difference between 4 and 10. For all selected actions, we compute their reverse state-specific effect together with its variance. We keep those with absolute r-SSE $\geq 0.1$ and variance $\geq 0.01$, which yields a total of 437 alternative actions for the evaluation of r-SSE-ICC.

For each selected action and its reverse state-specific effect, we compute the contribution scores assigned to all state

variables by the r-SSE-ICC method. We are interested in seeing how spread the scores are across the states, i.e., if our method attributes the effect equally or if it assigns larger scores to few states. To achieve this, we depict the Gini coefficient distribution (Gini, 1936) of these scores for various round differences in Plot 3c.[4] Our results indicate that, independently of the trajectory size, r-SSE-ICC pinpoints for most trajectories only a small subset of state variables with significant (intrinsic) contribution to the reverse state-specific effect. In practice, this means that for this setting we actually need to infer the counterfactual values of only a few key states in each trajectory in order to accurately estimate r-SSE. This is an interesting observation, as it implies that given a set of trajectories, r-SSE-ICC can reveal aspects of the underlying counterfactual distribution.

## 7. Discussion

In this paper, we introduce a causal explanation framework tailored to multi-agent MDPs. Specifically, we decompose the total counterfactual effect of an agent's action by attributing it to the agents' behavior or environment dynamics. Our experimental results demonstrate that our decomposition provides valuable insights into the distinct roles that agents and environment play in influencing the effect. To the best of our knowledge, this is the first work that looks into the problem of counterfactual effect decomposition in the context of multi-agent sequential decision making. While our findings are promising, there are several directions for future exploration, which we outline below.

**Computational complexity.** The computational complexity of our decomposition approach mainly depends on the

---

[4]In measuring the Gini coefficient we consider only the state variables that follow the intervention, and correspond to time-steps at which it was the AI's turn to take action. These are the only states that can be attributed a non-zero contribution by r-SSE-ICC.

total number of agents and the length of the MMDP's time horizon. In our experiments, we use a relatively small number of agents and a horizon of a few dozen time-steps. We believe that many interesting multi-agent settings belong to this regime, e.g., human-AI collaboration. Nevertheless, there are settings in which computational complexity considerations can be important, and we see this as an interesting future research direction to explore. In Appendix I, we analyze the computational complexity of the ASE-SV and r-SSE-ICC methods, and discuss potential mitigation strategies for when the number of agents or the time horizon are prohibitively large. We also discuss how agents' capabilities can affect the computational complexity of our approach.

**Causal assumptions.** Making causal assumptions in order to enable counterfactual identifiability is quite common in the literature. There is a plethora of works at the intersection of decision making and counterfactual reasoning that assumes exogeneity alongside causal properties, such as weak (Triantafyllou et al., 2024) or strong (Tsirtsis & Rodriguez, 2024) noise monotonicity, counterfactual stability (Oberst & Sontag, 2019), or access to the ground-truth causal model (Richens et al., 2022). However, these assumptions are often violated in practice. Extending the applicability of our proposed approach to non-identifiable domains would therefore be of significant practical value. For instance, our effect decomposition approach could utilize bounds on counterfactual probabilities instead of relying on point estimates, albeit on the expense of *efficiency*. Such bounds can be obtained from observational data through *partial identification* methods (Manski, 1990; Zhang et al., 2022).

**Applications to accountable decision making.** We deem the problem of decomposing counterfactual effects particularly relevant for multi-agent decision-making settings where accountability is paramount. Our approach can be applied in these settings, by integrating it into existing causal tools for retrospectively analyzing decision-making failures. For instance, consider methods for *blame attribution* in multi-agent systems (Halpern & Kleiman-Weiner, 2018; Friedenberg & Halpern, 2019). Typically, these methods first identify the agents' actions that were critical to the outcome, i.e., those that, had they been different, would have likely prevented failure. Next, they assess the agents' *epistemic states*, determining to what extent each agent could or should have predicted the consequences of acting differently. Our approach can enhance these methods by offering a more granular notion of blame. In the Sepsis scenario described in Section 1, for example, the clinician may be expected to predict how their actions directly affect the patient's state, but may not be expected to predict the AI's responses, especially if they have never worked with the current version of the model before. According to the output of our decomposition approach (Plot 1b), the clinician would then receive 73.5% of the total blame for their action, rather than bearing

full responsibility. We see significant potential in combining our approach with existing works on blame attribution and related concepts in accountable decision making, offering practical benefits across various multi-agent domains.

## Impact Statement

This paper aims to advance the field of Machine Learning, particularly in accountable multi-agent sequential decision-making. Our approach to decomposing counterfactual effects can be integrated into existing causal tools for retrospectively analyzing decision-making failures, offering deeper insights into accountability (see Section 7 for an example on blame attribution). While our work has potential societal implications, we do not find any that require specific emphasis here.

## Acknowledgements

This research was, in part, funded by the Deutsche Forschungsgemeinschaft (DFG, German Research Foundation) – project number 467367360.

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

# A. List of Appendices

In this section, we provide a brief description of the content provided in the appendices of the paper.

- Appendix B contains a table that summarizes the most important notation used in the paper.

- Appendix C provides additional information on MMDP-SCMs.

- Appendix D contains the causal graph of the MMDP-SCM from Section 2.2.

- Appendix E provides additional information on noise monotonicity.

- Appendix F formally states the properties defined in Section 5 for the ASE-SV method.

- Appendix G outlines an algorithm for approximating the conditional variance from Eq. 4.

- Appendix H contains the proofs of Theorems 3.3 and 4.2.

- Appendix I provides a discussion on the computational complexity of the ASE-SV and r-SSE-ICC methods.

- Appendix J provides additional information on the experimental setup and implementation details.

- Appendix K includes additional experimental results.

- Appendix L includes additional experiments assessing the estimation error of our empirical results.

- Appendix M includes additional experiments assessing the robustness of our empirical results to the noise monotonicity assumption.

- Appendix N provides a graphical illustration of all counterfactual effects introduced in Sections 2 and 3 using the Sepsis example from (Triantafyllou et al., 2024).

## B. Notation Summarization Table

Table 1: Summarizes the most important notation used in the paper.

| Notation | Meaning |
|---|---|
| $M$ | MMDP-SCM |
| $P(\cdot)_M$ | Probability defined over $M$ |
| $\{1, ..., n\}$ | Set of agents |
| $h$ | Time horizon |
| $S_t, s_t \in \mathcal{S}$ | State variable and value at time-step $t$ |
| $A_{i,t}, a_{i,t} \in \mathcal{A}_i$ | Action variable and value of agent $i$ at time-step $t$ |
| $\tau, \tau(X)$ | Trajectory and value of variable $X$ in $\tau$ |
| $P(\cdot\|\tau; M')_M$ | Probability conditioned on trajectory $\tau$ generated by $M'$ |
| $\mathbf{U}, \mathbf{u}$ | Vector of noise variables and vector of noise values |
| $P(\mathbf{u}), P(\mathbf{u}\|\tau)$ | Prior and posterior noise distributions |
| $do(A_{i,t} := a_{i,t})$ | Hard intervention on $A_{i,t}$ |
| $M^{do(A_{i,t}:=a_{i,t})}$ | Modified MMDP-SCM |
| $Y$ | Response/Outcome variable |
| $Y_{a_{i,t}}$ | Potential response of $Y$ to $do(A_{i,t} := a_{i,t})$ |
| $do(Y := Y_{a_{i,t}})$ | Natural intervention on $Y$ |
| $P(y_{a_{i,t}}\|\tau)_M$ | Counterfactual probability of $Y = y$ under $do(A_{i,t} := a_{i,t})$ |
| $\text{TCFE}_{a_{i,t},\tau(A_{i,t})}(Y\|\tau)_M$ | Definition 2.1: Total counterfactual effect (TCFE) |
| $I$ | A set of interventions on action variables |
| $\text{ASE}^{\{1,...,n\}}_{a_{i,t},\tau(A_{i,t})}(Y\|\tau)_M$ | Definition 3.1: Total agent-specific effect (tot-ASE) |
| $\text{SSE}_{a_{i,t},\tau(A_{i,t})}(Y\|\tau)_M$ | Definition 3.2: State-specific effect (SSE) |
| $\text{SSE}_{\tau(A_{i,t}),a_{i,t}}(Y\|\tau)_M$ | Equation 2: Reverse state-specific effect (r-SSE) |
| $\Delta Y_{I,a_{i,t}}$ | Difference in potential responses $Y_I - Y_{a_{i,t}}$ |
| $\text{ICC}(S_k \to \Delta Y_{I,a_{i,t}}\|\tau)$ | Equation 4: Intrinsic causal contribution (ICC) |
| $\psi_{S_k}$ | Score assigned to state $S_k$ by the r-SSE-ICC (Definition 4.1) |
| $\text{ASE}^{\mathbf{N}}_{a_{i,t},\tau(A_{i,t})}(Y\|\tau)_M$ | Definition 5.1: Agent-specific effect (ASE) |
| $\phi_i$ | Score assigned to agent $i$ by the ASE-SV (Definition 5.2) |

## C. Additional Information on MMDP-SCMs

Consider an MMDP-SCM $M = \langle \mathbf{V}, \mathbf{U}, P(\mathbf{u}), \mathcal{F} \rangle$. For the *observational distribution* of $M$, $P(\mathbf{V})$, to be consistent with an MMDP $\langle \mathcal{S}, \{1, ..., n\}, \mathcal{A}, T, h, \sigma \rangle$ and a joint policy $\pi$, functions in $\mathcal{F}$ and noise distribution $P(\mathbf{u})$ need to satisfy the following conditions for every $(s, \mathbf{a}, s')$ triplet and time-step $t$:

$$\int_{u^{S_0}:f^{S_0}(u^{S_0})=s} P(u^{S_0}) = P(S_0 = s|\sigma); \quad \int_{u^{S_t}:f^S(s,\mathbf{a},u^{S_t})=s'} P(u^{S_t}) = T(s'|s,\mathbf{a});$$

$$\int_{u^{A_{i,t}}:f^{A_i}(s,u^{A_{i,t}})=a_i} P(u^{A_{i,t}}) = \pi_i(a_i|s). \tag{5}$$

The first two conditions in Eq. 5 guarantee that $M$ induces the initial state distribution and state transition dynamics of the MMDP. The third condition makes sure that the action variables in $M$ agree with the joint policy $\pi$.

## D. Causal Graph of MMDP-SCM

This section contains the causal graph of the MMDP-SCM described in Section 2.2. The causal graph is shown in Fig. 4.

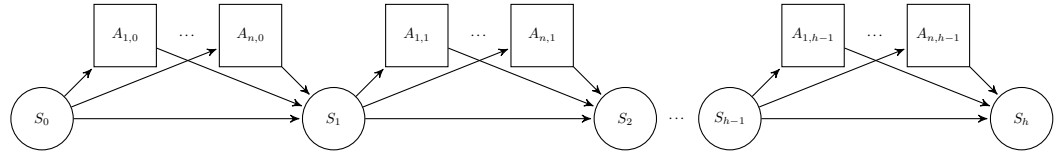

Figure 4: The causal graph of an MMDP-SCM with $n$ agents and horizon $h$. Exogenous variables are omitted.

## E. Additional Information on Noise Monotonicity

In this section, we define the (weak) noise monotonicity property for categorical SCMs. It has been shown that noise monotonicity enables counterfactual identifiability. For more details on noise monotonicity and its connection to the identifiability problem, we refer the interested reader to (Triantafyllou et al., 2024).

**Definition E.1** (Noise Monotonicity). Given an SCM $M$ with causal graph $G$, we say that variable $V^i \in \mathbf{V}$ is noise-monotonic in $M$ w.r.t. a total ordering $\leq_i$ on $\text{dom}\{V^i\}$, if for any $pa^i \in \text{dom}\{Pa^i(G)\}$ and $u_1^i, u_2^i \sim P(U^i)$ s.t. $u_1^i < u_2^i$, it holds that $f^i(pa^i, u_1^i) \leq_i f^i(pa^i, u_2^i)$.

Essentially, noise monotonicity assumes that all observed variables in an SCM, or MMDP-SCM in our paper, are monotonic w.r.t. their corresponding noise variable (for some specified total ordering). Note that noise monotonicity is not limiting for the MMDPs or agents' policies. In simple words, what noise monotonicity assumption restricts is the expressivity of counterfactual distributions. There can be many MMDP-SCMs whose observational distribution is consistent with the MMDP, but admit different counterfactual distributions. Theorem 4.3 in (Triantafyllou et al., 2024) shows that by limiting the class of possible MMDP-SCMs to the ones that satisfy noise monotonicity, counterfactual identifiability is guaranteed.

## F. Properties for ASE-SV

In this section, we formally state the properties defined in Section 5 for the ASE-SV method.

**Efficiency:** *The total sum of agents' contribution scores is equal to the total agent-specific effect.* Formally,

$$\sum_{j \in \{1,...,n\}} \phi_j = \text{ASE}_{a_{i,t}, \tau(A_{i,t})}^{\{1,...,n\}}(Y|\tau)_M.$$

**Invariance:** *Agents who do not marginally contribute to the total agent-specific effect are assigned a zero contribution score.* Formally, if for every $S \subseteq \{1,...,n\} \backslash \{j\}$

$$\text{ASE}_{a_{i,t}, \tau(A_{i,t})}^{S\cup\{j\}}(Y|\tau)_M - \text{ASE}_{a_{i,t}, \tau(A_{i,t})}^{S}(Y|\tau)_M = 0,$$

then $\phi_j = 0$.

**Symmetry:** *Agents who contribute equally to the total agent-specific effect are assigned the same contribution score.* Formally, if for every $S \subseteq \{1,...,n\} \backslash \{j,k\}$

$$\text{ASE}_{a_{i,t}, \tau(A_{i,t})}^{S\cup\{j\}}(Y|\tau)_M - \text{ASE}_{a_{i,t}, \tau(A_{i,t})}^{S}(Y|\tau)_M = \text{ASE}_{a_{i,t}, \tau(A_{i,t})}^{S\cup\{k\}}(Y|\tau)_M - \text{ASE}_{a_{i,t}, \tau(A_{i,t})}^{S}(Y|\tau)_M,$$

then $\phi_j = \phi_k$.

**Contribution monotonicity:** *The contribution score assigned to an agent depends only on its marginal contributions to the total agent-specific effect and monotonically so.* Formally, let $M_1$ and $M_2$ be two MMDP-SCMs with $n$ agents, if for every $S \subseteq \{1,...,n\} \backslash \{j\}$

$$\text{ASE}_{a_{i,t}, \tau(A_{i,t})}^{S\cup\{j\}}(Y|\tau)_{M_1} - \text{ASE}_{a_{i,t}, \tau(A_{i,t})}^{S}(Y|\tau)_{M_1} \geq \text{ASE}_{a_{i,t}, \tau(A_{i,t})}^{S\cup\{j\}}(Y|\tau)_{M_2} - \text{ASE}_{a_{i,t}, \tau(A_{i,t})}^{S}(Y|\tau)_{M_2},$$

then $\phi_j^{M_1} \geq \phi_j^{M_2}$.

# G. Algorithm for Conditional Variance

In this section, we present our approach for approximating the expected conditional variance from Eq. 4. Algorithm 1 estimates $\mathbb{E}[\mathrm{Var}(\Delta Y_{I,a_{i,t}}|\tau, \mathbf{U}^{<S_k})|\tau]_M$. To estimate the conditional variance $\mathbb{E}[\mathrm{Var}(\Delta Y_{I,a_{i,t}}|\tau, \mathbf{U}^{<S_k}, \mathbf{U}^{S_k})|\tau]_M$, it suffices to modify Algorithm 1 to sampling conditioning noise variables from $P(\mathbf{u}^{<S_k}, \mathbf{u}^{S_k}|\tau)$ and non-conditioning ones from $P(\mathbf{u}^{\geq S_{k+1}}|\tau)$.

---

**Algorithm 1** Estimates $\mathbb{E}[\mathrm{Var}(\Delta Y_{I,a_{i,t}}|\tau, \mathbf{U}^{<S_k})|\tau]_M$

---

**Input**: MMDP-SCM $M$, trajectory $\tau$, action variable $A_{i,t}$, action $a_{i,t}$, response variable $Y$, state variable $S_k$, number of conditioning/non-conditioning posterior samples $H_1/H_2$

1: $h_1 \leftarrow 0, h_2 \leftarrow 0$
2: $\mu_1 \leftarrow 0, \mu_2 \leftarrow 0$
3: **while** $h_1 < H_1$ **do**
4: $\quad$ $\mathbf{u}_{\mathrm{cond}} \sim P(\mathbf{u}^{<S_k}|\tau)$ $\quad$ # Sample conditioning noise variables
5: $\quad$ $h_1 \leftarrow h_1 + 1$
6: $\quad$ $c_1 \leftarrow 0, c_2 \leftarrow 0$
7: $\quad$ **while** $h_2 < H_2$ **do**
8: $\quad\quad$ $\mathbf{u}_{\mathrm{non}} \sim P(\mathbf{u}^{\geq S_k}|\tau)$ $\quad$ # Sample non-conditioning noise variables
9: $\quad\quad$ $h_2 \leftarrow h_2 + 1$
10: $\quad\quad$ $\mathbf{u} = (\mathbf{u}_{\mathrm{cond}}, \mathbf{u}_{\mathrm{non}})$
11: $\quad\quad$ $\tau^{\mathrm{cf}} \sim P(\mathbf{V}|\mathbf{u})_{M^{do(A_{i,t}:=a_{i,t})}}$ $\quad$ # Compute counterfactual trajectory
12: $\quad\quad$ $y^{\mathrm{cf}} \leftarrow \tau^{\mathrm{cf}}(Y)$
13: $\quad\quad$ $I \leftarrow \{A_{i',t'} := \tau^{\mathrm{cf}}(A_{i',t'})\}_{i'\in\{1,...,n\},t'>t}$
14: $\quad\quad$ $y^I \sim P(Y|\mathbf{u})_{M^{do(I)}}$ $\quad$ # Compute response to natural intervention
15: $\quad\quad$ $c_1 \leftarrow c_1 + (y^I - y^{\mathrm{cf}})$
16: $\quad\quad$ $c_2 \leftarrow c_2 + (y^I - y^{\mathrm{cf}})^2$
17: $\quad$ **end while**
18: $\quad$ $\mu_1 \leftarrow \mu_1 + \left(\frac{c_1}{H_2}\right)^2$
19: $\quad$ $\mu_2 \leftarrow \mu_2 + \frac{c_2}{H_2}$
20: $\quad$ $h_2 \leftarrow 0$
21: **end while**
22: **return** $\frac{\mu_2 - \mu_1}{H_1}$

---

# H. Proofs

## H.1. Proof of Theorem 3.3

*Proof.* Eq. 3 follows directly from Definition 2.1, Definition 3.1 and Eq. 2:

$$
\begin{aligned}
\mathrm{TCFE}_{a_{i,t},\tau(A_{i,t})}(Y|\tau)_M &= \mathbb{E}[Y_{a_{i,t}}|\tau]_M - \tau(Y) \\
&= \mathbb{E}[Y_{a_{i,t}}|\tau]_M - \tau(Y) + \mathbb{E}[Y|\tau;M]_{M^{do(I)}} - \mathbb{E}[Y|\tau;M]_{M^{do(I)}} \\
&= \mathrm{ASE}_{a_{i,t},\tau(A_{i,t})}^{\{1,...,n\}}(Y|\tau)_M - \mathrm{SSE}_{\tau(A_{i,t}),a_{i,t}}(Y|\tau)_M,
\end{aligned}
$$

where $I = \{A_{i',t'} := A_{i',t'[a_{i,t}]}\}_{i'\in\{1,...,n\},t'>t}$.

$\square$

## H.2. Proof of Theorem 4.2

*Proof.* It holds that

$$
\sum_{k \in [t+1, t_Y]} \psi_{S_k} = \frac{\mathbb{E}[\mathrm{Var}(\Delta Y_{I,a_{i,t}} | \tau, \mathbf{U}^{<S_{t+1}}) | \tau]_M - \mathbb{E}[\mathrm{Var}(\Delta Y_{I,a_{i,t}} | \tau, \mathbf{U}^{<S_{t_Y}}, \mathbf{U}^{S_{t_Y}}) | \tau]_M}{\mathrm{Var}(\Delta Y_{I,a_{i,t}} | \tau)} \cdot \mathrm{SSE}_{\tau(A_{i,t}), a_{i,t}}(Y | \tau)_M
$$

$$
= \frac{\mathbb{E}[\mathrm{Var}(\Delta Y_{I,a_{i,t}} | \tau, \mathbf{U}^{<S_{t+1}}) | \tau]_M - \mathbb{E}[\mathrm{Var}(\Delta Y_{I,a_{i,t}} | \tau, \mathbf{U}) | \tau]_M}{\mathrm{Var}(\Delta Y_{I,a_{i,t}} | \tau)} \cdot \mathrm{SSE}_{\tau(A_{i,t}), a_{i,t}}(Y | \tau)_M
$$

$$
= \frac{\mathbb{E}[\mathrm{Var}(\Delta Y_{I,a_{i,t}} | \tau, \mathbf{U}^{<S_{t+1}}) | \tau]_M}{\mathrm{Var}(\Delta Y_{I,a_{i,t}} | \tau)} \cdot \mathrm{SSE}_{\tau(A_{i,t}), a_{i,t}}(Y | \tau)_M.
$$

First step holds because $\mathbb{E}[\mathrm{Var}(\Delta Y_{I,a_{i,t}} | \tau, \mathbf{U}^{<S_k}, \mathbf{U}^{S_k}) | \tau]_M = \mathbb{E}[\mathrm{Var}(\Delta Y_{I,a_{i,t}} | \tau, \mathbf{U}^{<S_{k+1}}) | \tau]_M$, for every $k \in \{t + 1, ..., t_Y - 1\}$. The second step follows from the fact that noise terms associated with observed variables which (chronologically) proceed $t_Y$ do not influence the value of $\Delta Y_{I,a_{i,t}}$. The third step holds because the expected conditional variance satisfies *calibration*, i.e., $\mathbb{E}[\mathrm{Var}(\Delta Y_{I,a_{i,t}} | \tau, \mathbf{U}) | \tau]_M = 0$.

Let now $X$ be any ancestor of $S_{t+1}$ in the causal graph of $M$, apart from $A_{i,t}$. Note that $X$ is not affected by interventions $do(I)$ and $do(A_{i,t} := a_{i,t})$. Therefore, the solution of $X$ in the MMDP-SCMs $M^{do(I)}$ and $M^{do(A_{i,t} := a_{i,t})}$ will be equal to its factual value in $\tau$, i.e, $\tau(X)$, for every context $\mathbf{u}$ sampled from the posterior $P(\mathbf{u} | \tau)$. Furthermore, $A_{i,t}$ is fixed to $a_{i,t}$ in $M^{do(A_{i,t} := a_{i,t})}$, while it is also not affected by $do(I)$. It follows that conditioning on the noise terms associated with $X$ or $A_{i,t}$ does not reduce the variance of $\Delta Y_{I,a_{i,t}}$. Therefore, it holds that

$$
\mathbb{E}[\mathrm{Var}(\Delta Y_{I,a_{i,t}} | \tau, \mathbf{U}^{<S_{t+1}}) | \tau]_M = \mathrm{Var}(\Delta Y_{I,a_{i,t}} | \tau),
$$

which concludes our proof.

$\square$

# I. Discussion on Computational Complexity

In this section, we analyze the computational complexity of the ASE-SV (Definition 5.2) and r-SSE-ICC (Definition 4.1) methods, and discuss potential mitigation strategies for when the number of agents or the length of the time horizon are prohibitively large. We conclude the section with a discussion about the effect of agents' capabilities on the computational complexity of our approach.

## I.1. Computational Complexity of ASE-SV

The number of agent-specific effect evaluations required by the exact ASE-SV calculation grows exponentially with the number of agents $n$. One potential mitigation strategy for this problem is to adapt to our setting sampling based approaches that efficiently approximate Shapley value without violating *efficiency*, i.e., attributing the entire effect. (Jia et al., 2019) propose such an algorithmic approach, which requires $O(n(\log n)^2)$ evaluations for any bounded utility. This means that their algorithm is applicable to ASE-SV in settings where the value of agent-specific effects is bounded, as is the case in both our experiments.

## I.2. Computational Complexity of r-SSE-ICC

Computing the contribution scores assigned by the r-SSE-ICC method to **all** state variables requires $O(h)$, where $h$ denotes the time horizon, executions of Algorithm 1. When we deal with long-horizon MMDPs, this linear dependence on the number of time-steps can slow down our method. One intuitive strategy to reduce the number of computations in this case is by grouping together state variables from consecutive time-steps. That way, the r-SSE-ICC method would attribute the effect to sets of consecutive state variables instead of individual ones. If the time horizon (between action and outcome) is partitioned in groups of the same fixed size $k$, except maybe for the last one, then the modified r-SSE-ICC method would require $O(\frac{h}{k})$ executions of Algorithm 1.

In settings where it is reasonable to assume or empirically verify that r-SSE-ICC is sparse, in the sense that only a few state variables have significant (intrinsic) contributions to the effect, as it is the case in both our experiments,

then we are able to further reduce the number of Algorithm 1 executions. More specifically, we can utilize the fact that the expected noise-conditional variance measure satisfies *monotonicity*, i.e., $E[\text{Var}(\Delta Y_{I,a_{i,t}}|\tau, \mathbf{U}^{<S_k})|\tau]_M \geq E[\text{Var}(\Delta Y_{I,a_{i,t}}|\tau, \mathbf{U}^{<S_k}, \mathbf{U}^{S_k})|\tau]_M$. If we know, for example, that for most of the times there is at most one state variable with non-negligible (intrinsic) contribution to the effect, we can simply use a binary search approach to pinpoint that state. This can reduce the complexity of r-SSE-ICC to $O(\log(h))$ executions.

### I.3. Effect of Agents' Capabilities on Computational Complexity

The complexity of decision-making agents affects the computational complexity of our decomposition approach, assuming that increased capabilities imply increased inference time. The reasoning is the following: our approach to estimating counterfactual effects involves sampling trajectories from the posterior distribution and then averaging the values of the response variable across these trajectories. Sampling a trajectory from the posterior distribution generally requires to prompt each agent once for every counterfactual state in which they need to act.

For reference, in the Gridworld environment, more than $90\%$ of the time required to sample one counterfactual trajectory is spent on the inference of the LLM agent, while the remaining $\sim 10\%$ is shared between the two RL agents.[5] Consequently, if we were to use an LLM agent with reduced cognitive capabilities, and hence less inference time, then the scalability of our approach in this experiment would significantly improve.

## J. Experimental Setup and Implementation

In this section, we provide additional information on the experimental setup and implementation.

### J.1. Gridworld

**Setup.** Our setup is an adaptation of the **Planner-Actor-Reporter** system from (Dasgupta et al., 2023). The Planner is tasked with understanding the high-level steps necessary for the completion of a task and then breaking it down to a sequence of instructions. Actors are RL agents pre-trained to complete a set of simple instructions in the environment. Lastly, the Reporter is tasked with translating environment observations into a textual representation comprehensible by the Planner.

**Environment.** We consider the gridworld environment depicted in Fig. 2a with two actors, $\mathcal{A}_1$ and $\mathcal{A}_2$. There are two *boxes* located on the rightmost corners, each of which contains two objects. Each object has a color that determines its value, in particular, pink > green > yellow. The object's color is randomly sampled at the beginning of each trajectory. Objects can be picked up and carried by the actors – each actor can pick up only one object, and only one object can be picked up from each box. Grey-colored cells represent *walls*. Blank cells indicate areas of small negative cost. Colored cells indicate areas of larger stochastic penalty, which is significantly reduced when actors carry an object of a matching color. Penalties induced by cells of the same color share the same means, but might differ in their underlying distributions. Moreover, in expectation, pink cells inflict higher penalties than green ones, and green cells higher than yellow ones. Cells denoted with stars are delivery locations. If an object is delivered to the location with the matching color, then the object's value is rewarded. The objective in this environment is to maximize the combined total return of both actors. The full reward specification can be found in Table 2.

**Instructions.** The Gridworld environment supports a simplified set of 8 instructions: *examine box 1*, *examine box 2*, *pickup pink*, *pickup green*, *pickup yellow*, *goto pink*, *goto green* and, *goto yellow*. We pre-train both actors to learn a goal-conditioned policy for executing each of the available instructions. During training, we sample a new instruction at the beginning of each trajectory. Additionally, we initialize an actor according to the instruction and randomize over its valid observation space. For example, for the instruction *goto pink*, we initialize the actor to its respective position (under/above the first/second box for $\mathcal{A}_1$ and $\mathcal{A}_2$ respectively) and randomly select the object it's carrying. The actor is rewarded positively whenever it completes the instruction.

**Actors.** Actors $\mathcal{A}_1$ and $\mathcal{A}_2$ spawn on the same fixed locations at the beginning of each trajectory. Apart from movement actions, actors can also perform *pickup* actions when located next to a box. The policies are represented via neural network parameters and are learned using double deep Q-learning (Mnih et al., 2015; Van Hasselt et al., 2016). Both agents take as their input concatenated, one-hot encoded vectors, which include their instruction, their current position and the color of the

---

[5]Raw values and additional details on the time compute of our experiments are included on the README file of our code, which can be found at https://github.com/stelios30/cf-effect-decomposition.git.

Table 2: **Reward specification for Gridworld.** An empty distribution column implies a deterministic reward issued upon entering the cell. For the green corridor, penalties are specified on a per-cell basis, identified by their zero-based indices into the associated row and column.

| Cell | Values | Distribution |
|---|---|---|
| All | -0.2 | - |
| Pink Penalty | [-30, -50, -70] | [1/3, 1/3, 1/3] |
| Pink Penalty (Reduced) | [-5, -15, -25] | [1/3, 1/3, 1/3] |
| Pink Delivery | +180 | - |
| Green Penalty $C_{2,4}$ | [-30, -40, -50] | [0.3, 0.4, 0.3] |
| Green Penalty $C_{2,4}$ (Reduced) | [-5, -10, -15] | [0.3, 0.4, 0.3] |
| Green Penalty $C_{2,5}$ | [-30, -40, -50] | [0.25, 0.5, 0.25] |
| Green Penalty $C_{2,5}$ (Reduced) | [-5, -10, -15] | [0.25, 0.5, 0.25] |
| Green Penalty $C_{2,6}$ | [-30, -40, -50] | [0.2, 0.6, 0.2] |
| Green Penalty $C_{2,6}$ (Reduced) | [-5, -10, -15] | [0.2, 0.6, 0.2] |
| Green Penalty $C_{2,7}$ | [-30, -40, -50] | [0.15, 0.7, 0.15] |
| Green Penalty $C_{2,7}$ (Reduced) | [-5, -10, -15] | [0.15, 0.7, 0.15] |
| Green Delivery | +150 | - |
| Yellow Penalty | [-25, -30, -35] | [1/3, 1/3, 1/3] |
| Yellow Penalty (Reduced) | [-2.5, -5, -7.5] | [1/3, 1/3, 1/3] |
| Yellow Delivery | +90 | - |

object they are carrying. We provide a full list of hyperparameters in Table 3. The hyperparameter optimization method was performed by randomly sampling 50 candidates from the specified ranges and selecting the combination that yielded the best test reward, averaged over all instructions.

**Planner and Reporter.** Planner is implemented using a pre-trained LLama 2.7B model (Touvron et al., 2023) and few-shot learning, to provide actors with instructions. More specifically, Planner can instruct actors to: *examine* a box, *pickup* an object and *deliver* that object to a specific destination. Furthermore, we assume an optimal Reporter whose task is to report to the Planner the necessary information about the state of the environment. In particular, the Reporter provides information about the boxes' contents and which objects were picked up by the actors (see Trajectories 1 and 2 for an illustrative example).

### J.2. Sepsis

Our experimental setup and implementation closely follow that of (Triantafyllou et al., 2024).

### J.3. Compute Resource

All experiments were run on a 64bit Debian-based machine having 2x12 CPU cores clocked at 3GHz with access to 1 TB of DDR3 1600MHz RAM and an NVIDIA A40 GPU. The software stack relied on Python 3.9.13, with installed standard scientific packages for numeric calculations and visualization (we provide a full list of dependencies and their exact versions as part of our code).

Table 3: Hyperparameters used for the Gridworld actors' policies.

| Parameter name | Parameter value | Tuning Range |
|---|---|---|
| Discount | 0.99 | [0.99, 0.9, 0.8] |
| Target Update Freq. | 1000 | [500, 1000, 1500] |
| Batch size | 512 | [256, 512, 1024, 2048] |
| Hidden Dim | 128 | [64, 128, 256] |
| Hidden Depth | 3 | [2, 3] |
| Learning Rate | 1e-4 | [1e-5, 5e-5, 1e-4, 5e-4, 1e-3] |
| Num. Estimation Step | 1 | [1, 3, 5, 10, 15] |

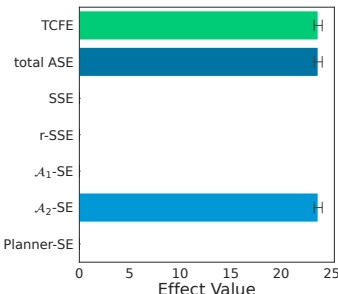

(a) Counterfactual Effects

Figure 5: Gridworld: Plot 5a shows the values of various counterfactual effects computed on the trajectory's total collected reward for the case when we intervene on the Planner's action at Step 2, forcing it to instruct $\mathcal{A}_2$ to pick up the green object instead of the pink one. Averages and standard errors are reported for 5 different seeds.

## K. Additional Experimental Results

### K.1. Gridworld

**Additional experiment.** We repeat the experiment from Section 6.1, but instead of intervening on $\mathcal{A}_2$'s pickup action we intervene on the Planner's action. In particular, we intervene on the Planner's action at Step 2, forcing it to instruct $\mathcal{A}_2$ to pick up the green object instead of the pink one. The total counterfactual effect of this intervention is equal to that of the intervention on $\mathcal{A}_2$'s action. However, the result of our decomposition approach for these two effects is different.

According to Plot 5a, the TCFE in this scenario is fully attributed to how the agents would respond to the intervention, and more specifically to the response of agent $\mathcal{A}_2$. Both the SSE and the r-SSE in this scenario are zero. This result is intuitive, as the Planner is not able to influence the state transitions directly, and hence the effect of its actions do not propagate through the environment dynamics. In contrast, the actions of $\mathcal{A}_2$ can influence the outcome through both the environment and future agent actions, and hence the decomposition of their effect is more nuanced (see Plot 2b).

**Trajectories.** We provide a textual depiction of the factual (Trajectory 1) and counterfactual (Trajectory 2) trajectories from Fig. 2a. We also provide a textual depiction of the counterfactual trajectory from the experiment described above (Trajectory 3).

### K.2. Sepsis

Fig. 6 illustrates the distribution of the r-SSE contribution scores computed by the r-SSE-ICC method in Section 6.2. Due to the use of a finite number of samples for estimating these scores, some negative values may be erroneously attributed. These cases have been excluded from the plots.

**Gridworld Trajectory 1 : Factual**

Box 1: (PINK, YELLOW); Box 2: (PINK, GREEN)

Step: 0; Reporter: $A_1$ respawn; $A_2$ respawn;
   Planner: (examine box 1, examine box 2); Reward 0.0;

Step: 1; Actors $(A_1, A_2)$: up, down; Reward: $-0.4$ $(A_1: -0.2, A_2: -0.2)$;

Step: 2; Reporter: $A_1$ (PINK YELLOW); $A_2$ (PINK GREEN);
   Planner: (pickup pink, pickup pink); Reward 0.0;

Step: 3; Actors $(A_1, A_2)$: pickup pink, pickup pink; Reward: $-0.4$ $(A_1: -0.2, A_2: -0.2)$;

Step: 4; Reporter: $A_1$ has PINK; $A_2$ has PINK;
   Planner: (goto pink, goto pink); Reward 0.0;

Step: 5; Actors $(A_1, A_2)$: down, up; Reward: $-0.4$ $(A_1: -0.2, A_2: -0.2)$;
Step: 6; Actors $(A_1, A_2)$: left, up; Reward: $-0.4$ $(A_1: -0.2, A_2: -0.2)$;
Step: 7; Actors $(A_1, A_2)$: left, up; Reward: $-0.4$ $(A_1: -0.2, A_2: -0.2)$;
Step: 8; Actors $(A_1, A_2)$: left, left; Reward: $-0.4$ $(A_1: -0.2, A_2: -0.2)$;
Step: 9; Actors $(A_1, A_2)$: left, left; Reward: $-0.4$ $(A_1: -0.2, A_2: -0.2)$;
Step: 10; Actors $(A_1, A_2)$: left, left; Reward: $-25.4$ $(A_1: -25.2, A_2: -0.2)$;
Step: 11; Actors $(A_1, A_2)$: left, left; Reward: $-25.4$ $(A_1: -25.2, A_2: -0.2)$;
Step: 12; Actors $(A_1, A_2)$: left, left; Reward: $-40.4$ $(A_1: -15.2, A_2: -25.2)$;
Step: 13; Actors $(A_1, A_2)$: left, left; Reward: $-30.4$ $(A_1: -5.2, A_2: -25.2)$;
Step: 14; Actors $(A_1, A_2)$: up, left; Reward: $-25.4$ $(A_1: -0.2, A_2: -25.2)$;
Step: 15; Actors $(A_1, A_2)$: left, left; Reward: $-25.4$ $(A_1: -0.2, A_2: -25.2)$;
Step: 16; Actors $(A_1, A_2)$: left, left; Reward: $-0.4$ $(A_1: -0.2, A_2: -0.2)$;
Step: 17; Actors $(A_1, A_2)$: left, up; Reward: $-0.4$ $(A_1: -0.2, A_2: -0.2)$;
Step: 18; Actors $(A_1, A_2)$: NULL, left; Reward: $-0.4$ $(A_1: -0.2, A_2: -0.2)$;
Step: 19; Actors $(A_1, A_2)$: NULL, left; Reward: $-0.4$ $(A_1: -0.2, A_2: -0.2)$;

Step: 20; Goal Reward: 360.0; Total Reward: 183.2;

---

**Gridworld Trajectory 2 : Counterfactual ($\mathcal{A}_2$'s action)**

---

Box 1: (PINK, YELLOW); Box 2: (PINK, GREEN)

Step: 0; Reporter: $A_1$ respawn; $A_2$ respawn;
      Planner: (examine box 1, examine box 2); Reward 0.0;

Step: 1; Actors ($A_1$, $A_2$): up, down; Reward: $-0.4$ ($A_1$: $-0.2$, $A_2$: $-0.2$);

Step: 2; Reporter: $A_1$ (PINK YELLOW); $A_2$ (PINK GREEN);
      Planner: (pickup pink, pickup pink); Reward 0.0;

Step: 3; Actors ($A_1$, $A_2$): pickup pink, pickup green; Reward: $-0.4$ ($A_1$: $-0.2$, $A_2$: $-0.2$);

Step: 4; Reporter: $A_1$ has PINK; $A_2$ has GREEN;
      Planner: (goto pink, goto green); Reward 0.0;

Step: 5; Actors ($A_1$, $A_2$): down, up; Reward: $-0.4$ ($A_1$: $-0.2$, $A_2$: $-0.2$);
Step: 6; Actors ($A_1$, $A_2$): left, left; Reward: $-0.4$ ($A_1$: $-0.2$, $A_2$: $-0.2$);
Step: 7; Actors ($A_1$, $A_2$): left, left; Reward: $-0.4$ ($A_1$: $-0.2$, $A_2$: $-0.2$);
Step: 8; Actors ($A_1$, $A_2$): left, up; Reward: $-0.4$ ($A_1$: $-0.2$, $A_2$: $-0.2$);
Step: 9; Actors ($A_1$, $A_2$): left, left; Reward: $-0.4$ ($A_1$: $-0.2$, $A_2$: $-0.2$);
Step: 10; Actors ($A_1$, $A_2$): left, left; Reward: $-25.4$ ($A_1$: $-25.2$, $A_2$: $-0.2$);
Step: 11; Actors ($A_1$, $A_2$): left, left; Reward: $-35.4$ ($A_1$: $-25.2$, $A_2$: $-10.2$);
Step: 12; Actors ($A_1$, $A_2$): left, left; Reward: $-30.4$ ($A_1$: $-15.5$, $A_2$: $-15.2$);
Step: 13; Actors ($A_1$, $A_2$): left, left; Reward: $-20.4$ ($A_1$: $-5.2$, $A_2$: $-15.2$);
Step: 14; Actors ($A_1$, $A_2$): up, left; Reward: $-15.4$ ($A_1$: $-0.2$, $A_2$: $-15.2$);
Step: 15; Actors ($A_1$, $A_2$): left, left; Reward: $-0.4$ ($A_1$: $-0.2$, $A_2$: $-0.2$);
Step: 16; Actors ($A_1$, $A_2$): left, left; Reward: $-0.4$ ($A_1$: $-0.2$, $A_2$: $-0.2$);
Step: 17; Actors ($A_1$, $A_2$): left, left; Reward: $-0.4$ ($A_1$: $-0.2$, $A_2$: $-0.2$);

Step: 18; Goal Reward: 330.0; Total Reward: 199.0;

---

## Gridworld Trajectory 3 : Counterfactual (Planner's action)

Box 1: (PINK, YELLOW); Box 2: (PINK, GREEN)

Step: 0; Reporter: $A_1$ respawn; $A_2$ respawn;
Planner: (examine box 1, examine box 2); Reward 0.0;

Step: 1; Actors ($A_1$, $A_2$): up, down; Reward: $-0.4$ ($A_1$: $-0.2$, $A_2$: $-0.2$);

Step: 2; Reporter: $A_1$ (PINK YELLOW); $A_2$ (PINK GREEN);
Planner: (pickup pink, pickup green); Reward 0.0;

Step: 3; Actors ($A_1$, $A_2$): pickup pink, pickup green; Reward: $-0.4$ ($A_1$: $-0.2$, $A_2$: $-0.2$);

Step: 4; Reporter: $A_1$ has PINK; $A_2$ has GREEN;
Planner: (goto pink, goto green); Reward 0.0;

Step: 5; Actors ($A_1$, $A_2$): down, up; Reward: $-0.4$ ($A_1$: $-0.2$, $A_2$: $-0.2$);
Step: 6; Actors ($A_1$, $A_2$): left, left; Reward: $-0.4$ ($A_1$: $-0.2$, $A_2$: $-0.2$);
Step: 7; Actors ($A_1$, $A_2$): left, left; Reward: $-0.4$ ($A_1$: $-0.2$, $A_2$: $-0.2$);
Step: 8; Actors ($A_1$, $A_2$): left, up; Reward: $-0.4$ ($A_1$: $-0.2$, $A_2$: $-0.2$);
Step: 9; Actors ($A_1$, $A_2$): left, left; Reward: $-0.4$ ($A_1$: $-0.2$, $A_2$: $-0.2$);
Step: 10; Actors ($A_1$, $A_2$): left, left; Reward: $-25.4$ ($A_1$: $-25.2$, $A_2$: $-0.2$);
Step: 11; Actors ($A_1$, $A_2$): left, left; Reward: $-35.4$ ($A_1$: $-25.2$, $A_2$: $-10.2$);
Step: 12; Actors ($A_1$, $A_2$): left, left; Reward: $-25.4$ ($A_1$: $-15.2$, $A_2$: $-10.2$);
Step: 13; Actors ($A_1$, $A_2$): left, left; Reward: $-15.4$ ($A_1$: $-5.2$, $A_2$: $-10.2$);
Step: 14; Actors ($A_1$, $A_2$): up, left; Reward: $-15.4$ ($A_1$: $-0.2$, $A_2$: $-15.2$);
Step: 15; Actors ($A_1$, $A_2$): left, left; Reward: $-0.4$ ($A_1$: $-0.2$, $A_2$: $-0.2$);
Step: 16; Actors ($A_1$, $A_2$): left, left; Reward: $-0.4$ ($A_1$: $-0.2$, $A_2$: $-0.2$);
Step: 17; Actors ($A_1$, $A_2$): left, left; Reward: $-0.4$ ($A_1$: $-0.2$, $A_2$: $-0.2$);

Step: 18; Goal Reward: 330.0; Total Reward: 209.0;

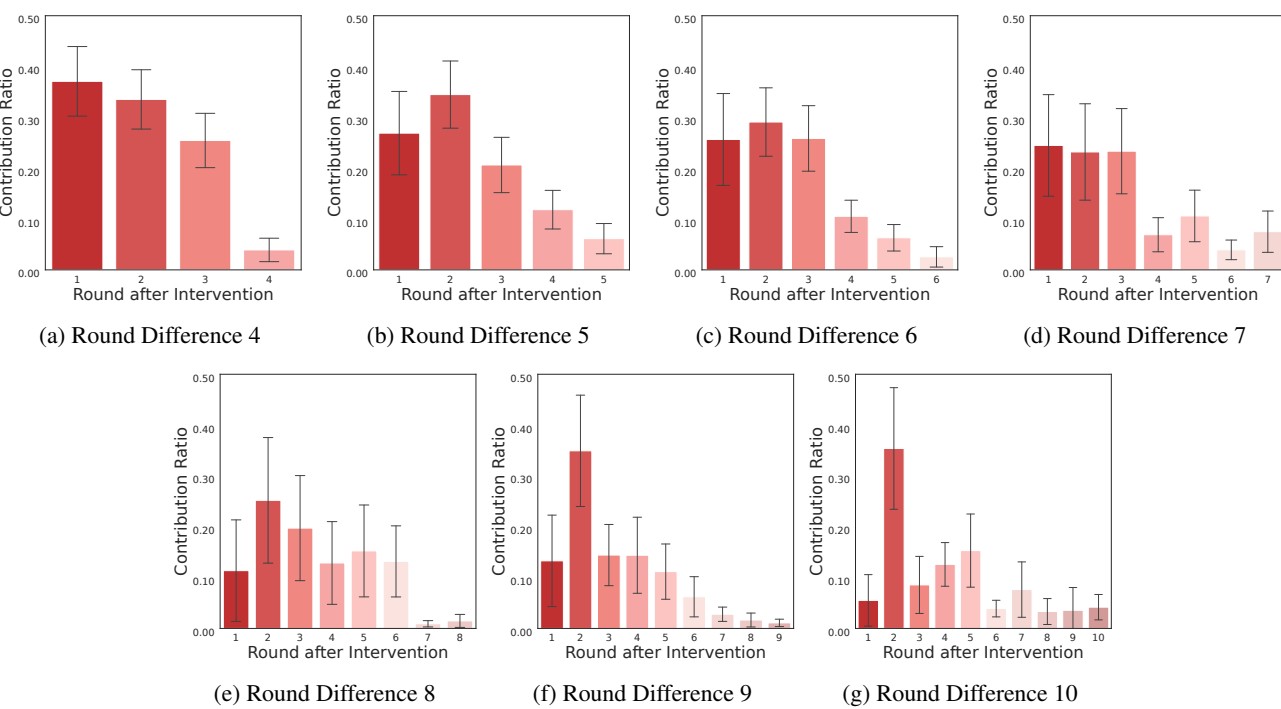

Figure 6: Sepsis: Plots 6a - 6g show the average contribution ratios attributed to the different state variables by the r-SSE-ICC method. Results are grouped based on the round difference of the selected actions (see Section 6.2 for an explanation). We plot the contributions only for state variables corresponding to rounds that follow the intervention All other contributions are zero. Averages and standard errors are reported for the 437 alternative actions chosen for the evaluation of r-SSE-ICC following the process described in Section 6.2.

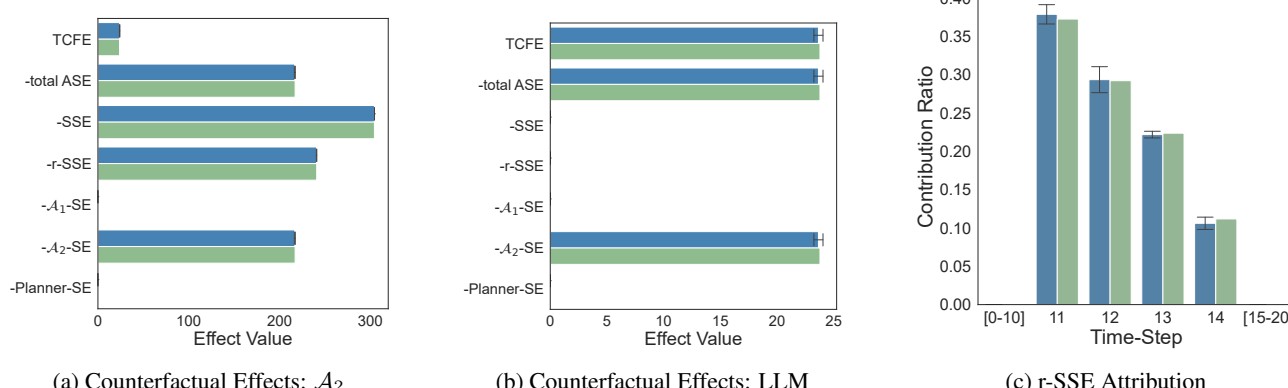

(a) Counterfactual Effects: $\mathcal{A}_2$    (b) Counterfactual Effects: LLM    (c) r-SSE Attribution

Figure 7: Plots 7a, 7b, and 7c replicate the empirical results from Plots 2b, 5a, and 2c, respectively, while additionally including the corresponding ground-truth values for all quantities. Estimated values from the original plots are shown in blue, while the ground-truth values are depicted in green.

## L. Additional Experiments Evaluating Estimation Error

To approximate counterfactual effects across all experiments presented in Section 6 and Appendix K, we employ posterior sampling-based methods akin to Algorithm 1. This is a standard approach to counterfactual inference ((Pearl, 2009)). In this section, we present additional experiments to evaluate the estimation error and support the reliability of our empirical findings.

### L.1. Gridworld

Fig. 7 reproduces Plots 2b, 2c and 5a, now including the ground-truth values of all estimated quantities for comparison. Notably, the ground-truth values (green) consistently lie within the standard error bounds of the estimated quantities (blue). The ground-truth counterfactual distribution for this experiment was obtained through direct computation.

### L.2. Sepsis

Compared to the Gridworld environment, acquiring ground-truth values for counterfactual quantities in the Sepsis setting is significantly more challenging. Instead, we analyze the standard error distributions by repeating the experiment across 10 different seeds. Specifically, for each alternative action selected for the evaluation of ASE-SV and r-SSE-ICC in Section 6.2, we perform the evaluation process 10 times and compute the empirical standard error for all estimated quantities: TCFE, tot-ASE, SSE, r-SSE, $\phi_{cl}$, $\phi_{ai}$, and r-SSE-ICC. Fig. 8 illustrates the resulting standard error distributions.[6]

The plots from Fig. 8 reveal minimal variability in the estimates of our causal explanation formula across seeds, with only a very small number of outliers. These results reinforce the reliability of our findings from Section 6.2 and support the robustness of our effect decomposition approach in the Sepsis experiment.

## M. Additional Experiments Evaluating Robustness to Noise Monotonicity

Throughout all experiments in this paper, we assume that noise monotonicity holds (see Appendix E for a formal definition) w.r.t. a chosen set of total orderings. In the Gridworld experiment, we design the environment such that penalty variables are noise-monotonic w.r.t. the numerical ordering – all other variables in this experiment are deterministic. In the Sepsis experiment, however, we lack access to the underlying causal model and rely solely on observational distributions. Consequently, the choice of total orderings for noise monotonicity in this experiment may influence the results. In this section, we present additional experiments to evaluate the robustness of the empirical findings from Section 6.2 to variations in the choice of the total orderings. We note that a similar evaluation was conducted in (Triantafyllou et al., 2024).

---

[6]We chose to plot standard error distributions grouped by the absolute average means of their estimates over other metrics of relative dispersion, such as Coefficient of Variation, due to the fact that for many of our estimates their mean value centers close to zero.

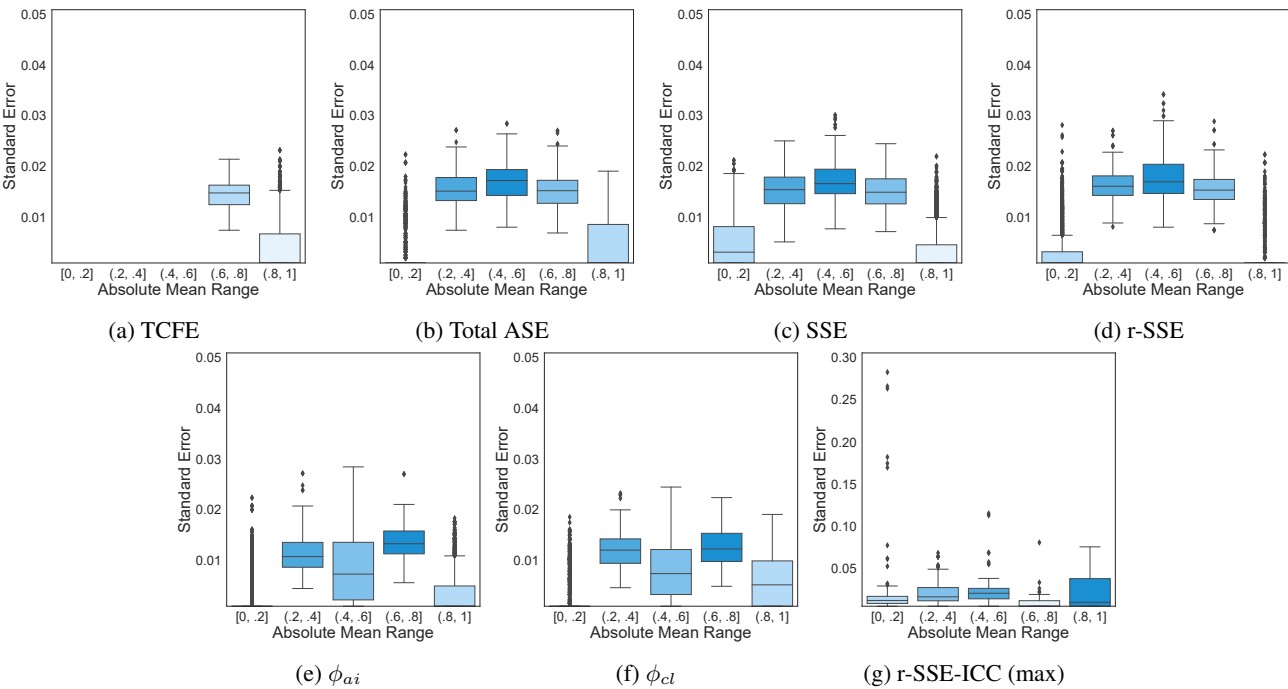

(a) TCFE  (b) Total ASE  (c) SSE  (d) r-SSE

(e) $\phi_{ai}$  (f) $\phi_{cl}$  (g) r-SSE-ICC (max)

Figure 8: Box Plots 8a-8f show the standard error distributions of all counterfactual estimates from Section 6.2 over all 8728 alternative actions selected for the evaluation of ASE-SV in that section. Box Plot 8g shows the standard error distribution of the scores assigned by the r-SSE-ICC method for a similar set of alternative actions as the one used in Section 6.2 for the evaluation of r-SSE-ICC. Among the multiple standard errors associated with each alternative action (one for each score assigned to a state variable), we report the one with the largest value. Standard errors and absolute mean values are measured across 10 different seeds.

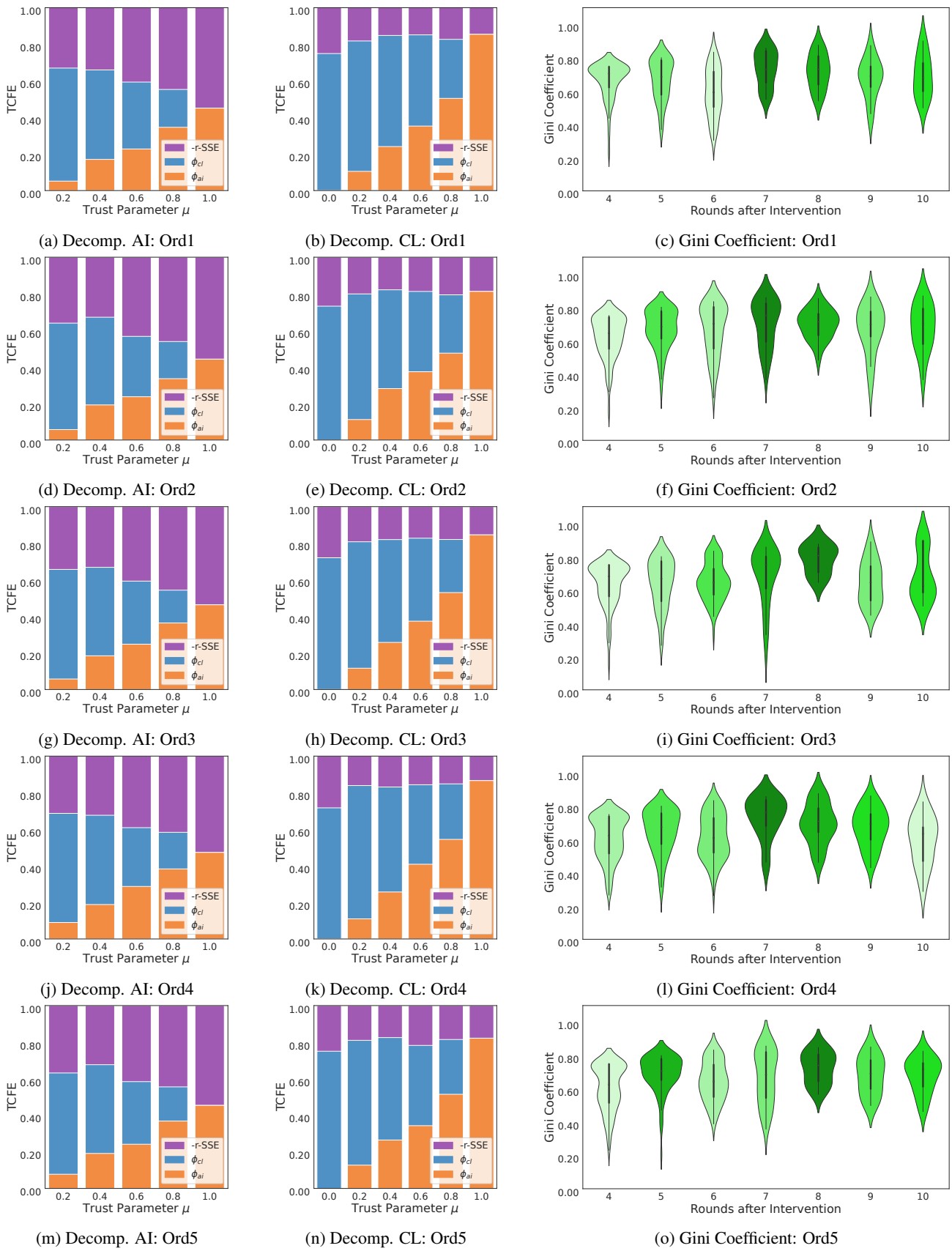

Figure 9: Repeats plots from Fig. 3 for 5 additional total orderings.

We repeat our experiments from Section 6.2 for 5 randomly selected additional total orderings. The results from these experiments are depicted in Fig. 9. From the plots corresponding to any of these total orderings, we can draw similar conclusions to the ones we drew from Fig. 3, especially from the plots that show the average percentage decomposition. We can conclude then that the empirical findings in Section 6.2 are robust to the uncertainty over the underlying total orderings.

## N. Graphical Illustration of Counterfactual Effects from Sections 2 and 3

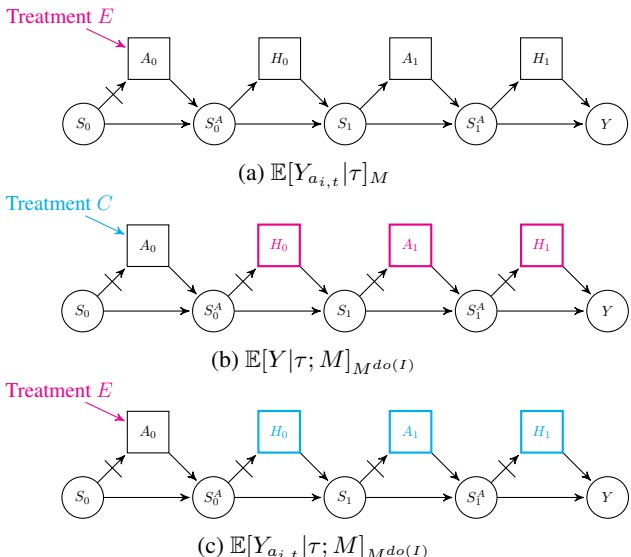

(a) $\mathbb{E}[Y_{a_{i,t}}|\tau]_M$

(b) $\mathbb{E}[Y|\tau; M]_{M^{do(I)}}$

(c) $\mathbb{E}[Y_{a_{i,t}}|\tau; M]_{M^{do(I)}}$

Figure 10: Depicts all counterfactual estimates appearing in Definitions 2.1 (TCFE), 3.1 (tot-ASE), 3.2 (SSE) and Equation 2 (r-SSE) using the Sepsis example from the introduction section of (Triantafyllou et al., 2024). The decision-making setting of this example is the same as the one from Section 1 and Section 6.2, but restricted to only two time-steps. We repeat the premise of the example and necessary notation for completeness. Squares in the graphs denote agents' actions, $A$ for AI and $H$ for clinician. Circles $S$ are patient states, while $S^A$ include both $S$ and $A$, i.e., $S^A = (S, A)$. $Y$ denotes the patient outcome after two time-steps. Edges that are striked through represent deactivated edges. Exogenous arrows represent interventions on $A_0$ that fix its value to one of two actions, Treatment $C$ or Treatment $E$. In the considered scenario, the former represents the action that was observed in the factual scenario ($\tau$), while the latter is the alternative treatment ($a_{i,t}$) whose counterfactual effect, on $Y$, we analyze. A cyan colored node signifies that the node is set to the action that the agent took in the factual scenario, i.e., under treatment $C$. A magenta colored node signifies that the node is set to the (counterfactual) action that the agent would have naturally taken under intervention $E$. Lastly, in Plot 10b $I = \{A_{i',t'} := A_{i',t'[a_{i,t}]}\}_{i' \in \{1,\ldots,n\}, t' > t}$ (Definition 3.1), while in Plot 10c $I = \{A_{i',t'} := A_{i',t'[\tau(A_{i',t'})]}\}_{i' \in \{1,\ldots,n\}, t' > t}$ (Definition 3.2).

