# OpenReview forum: "Counterfactual Effect Decomposition in Multi-Agent Sequential Decision Making"
_ICML.cc/2025/Conference — ICML 2025 poster_

### Official Review · Reviewer_n1oL · 2025-03-03

**Overall Recommendation:** 1

**Summary:**

This paper studies the attribution of counterfactual outcome to agent actions and states, that are total agent-specific effect and reverse state-specific effect. Moreover, it futher decompose the total agent-specific effect into individual agent effect, and the reverse state-specific
effect (r-SSE) into r-SSE-ICC. Through experiments on the Gridworld environment and sepsis management simulator, the interpretability of the method is demonstrated.

**Claims And Evidence:**

This paper try to explain the counterfactual effect of agent action. To achieve this, this paper defines a series of metrics such as tot-ASE. However, the definitions are complicated and not easy to understand. I observe the experimental results is mainly composed of measuring these metrics. I think the work does not accomplish the goal of interpretability.

**Essential References Not Discussed:**

I think the core references have been discussed and cited in the paper.

**Experimental Designs Or Analyses:**

I chech the experimental parts. The main concern is still that the results are composed of the proposed metrics which is not interpretable enough. I suggest authors to relate the interpretability result to some quantitative analysis which is easy to understand by human.

**Methods And Evaluation Criteria:**

The same to the previous part. The proposed method and evaluation criteria do not improve the interpretability of the agent actions.

**Other Comments Or Suggestions:**

In the last equation of Eq. (2), the difference of (1) \\(\\mathbb{E}[Y|\\tau; M]_{M^{do(I)}}\\) and the other term is not clear.

**Other Strengths And Weaknesses:**

The main strengths and weaknesses have been discussed in the previous parts.

**Questions For Authors:**

In Definition 5.2, the definition of $ASE^{S\cup \\{j\\}}$ is not clear, because the previous definition of ASE is designed for $\\{1,2,3,...,n\\}$

**Relation To Broader Scientific Literature:**

I think the contribution of the paper lies in the integration of causal media analysis and the multi-agent decision. Although it propose finer grained metrics of effect beyond Triantafyllou et al., 2024, I feel it does not help to better interpretability.

**Theoretical Claims:**

I chech the proof of Theorem 3.3, which I think is the cornerstone of the paper. However, I think the proof is wrong. The third equation does not relate the Definition 3.1 and Definition 3.2.

---

> ### Author Rebuttal · Authors · 2025-03-28
>
> Thank you for your valuable feedback. Please find below our response to your comments and questions.
>
> ## Response to Comments and Questions
>
> **Proof of Theorem 3.3.** We respectfully disagree with your concern regarding the correctness of the proof of Theorem 3.3. While we understand that the result may initially seem surprising, the proof follows in a straightforward manner. The first step is licensed by Definition 2.1, while the third step by Definition 3.1 and Equation (2). In the second step, we apply a standard add-and-subtract trick by introducing $\mathrm{E}[Y|\tau;M]_{M^{do(I)}}$ into the expression.
>
> **Interpretability.** Causal mediation analysis consists a well-established approach to interpreting the effect of an exposure variable to a response variable, by analyzing how it propagates through causal paths. Based on a similar principle, our proposed approach explains the effect of an agent's action on the outcome of an MMDP, by analyzing how it propagates through its influence on the agents’ behavior and the environment dynamics. Additionally, we have evaluated the interpretability of our decomposition approach by conducting extensive experiments on two RL environments featuring heterogeneous agents and diverse interaction protocols.
> The results of these evaluations indicate that our approach yields interpretable insights and aligns well with standard intuitions.
>
> Could we kindly ask the reviewer to elaborate on why they think our approach and evaluation protocol are not suitable for interpretability? If there are any specific concerns regarding our method, we would be glad to hear them and if possible try to also address them.
>
> **Equation (2).** Let $\tau$ be a trajectory generated by the MMDP-SCM $M$. Both $\mathrm{E}[Y_{a_{i,t}}|\tau]\_{M}$ and $\mathrm{E}[Y|\tau;M]\_{M^{do(I)}}$ measure the expected counterfactual value of outcome $Y$ in $\tau$, but under different interventions. The former corresponds to the intervention $do(A_{i,t} := a_{i,t})$, which fixes the action of agent $i$ at time-step $t$ to $a_{i,t}$. The latter corresponds to the intervention $do(I)$, which fixes all agents' actions, following time-step $t$, to the values that they would naturally take under intervention $do(A_{i,t} := a_{i,t})$.
>
> If it helps, we would like to kindly point out that in Section 6.1 of our paper, and particularly in the paragraph **Counterfactual effects**, you can find a detailed explanation of all introduced counterfactual measures, including r-SSE (Eq. 2), in the context of our Gridworld experiment. Furthermore, Appendix O provides a graphical illustration of all counterfactual estimates appearing in our paper using the Sepsis example from the introduction section of (Triantafyllou et al., 2024) and the causal graph used therein.
>
> **Definition 5.2.** $\text{ASE}^{\\{1, ..., n\\}}$ corresponds to the total agent-specific effect, which quantifies the effect of an intervention that propagates through all the agents. In contrast, $\text{ASE}^{S \cup \\{j\\}}$ quantifies the effect that propagates through only the agents included in the subset $S \cup \\{j\\} \subseteq \\{1, ..., n\\}$. Note that this distinction is explicitly highlighted in lines 283-290 (left column) of our manuscript.
>
> ## Conclusion
>
> We thank you again for your comments and questions. We would be happy to answer anything else in addition.

---

> > ### Comment · Reviewer_n1oL · 2025-04-07
> >
> > Thanks for your reply. Please give more details about the standard add-and-subtract trick in the proof of Theorem 3.3.

---

> > > ### Author Response · Authors · 2025-04-07
> > >
> > > Thank you for your follow-up. Please find our response below.
> > >
> > > In the second step of our proof, we are adding and subtracting the term $\mathrm{E}[Y|\tau;M]_{M^{do(I)}}$. Specifically, the expression
> > >
> > > $\mathrm{TCFE}\_{a_{i,t}, \tau(A_{i,t})}(Y|\tau)\_M = \mathrm{E}[Y_{a_{i,t}}|\tau]_M - \tau(Y)$
> > >
> > > is rewritten as
> > >
> > > $\mathrm{TCFE}\_{a_{i,t}, \tau(A_{i,t})}(Y|\tau)\_M = \mathrm{E}[Y_\{a_{i,t}}|\tau]\_M - \tau(Y) + \mathrm{E}[Y|\tau;M]\_{M^{do(I)}} - \mathrm{E}[Y|\tau;M]\_{M^{do(I)}}.$
> > >
> > > We hope this resolves any concerns about the proof’s validity.
> > >
> > > Please let us know if any other part remains unclear, we would be happy to further clarify. If all your concerns are addressed, we would also be grateful if you could consider updating your score.

---

### Official Review · Reviewer_LcKu · 2025-03-11

**Overall Recommendation:** 4

**Summary:**

The paper proposes a causal explanation formula for multi-agent Markov Decision Processes (MMDPs). It uses Structural Causal Models (SCMs) to decomposes the total counterfactual effect of an agent's action by attributing to each agent and state variable a score that results from their respective contributions to the outcome. Shapley values have been used  to attribute the total effect to individual agents.
Similarly, intrinsic causal contributions (ICC) have been used to decompose the reverse state-specific effect (r-SSE). Experiments were conducted on Gridworld environment with LLM-assisted agents and a sepsis management simulator.

## update after rebuttal:
I read the rebuttal of the authors as well as the questions of the reviewers. I maintain my score.

**Claims And Evidence:**

The main claim of the paper is that counterfactual effects of an agent's action in multi-agent sequential decision-making can be decomposed into agent-specific and state-specific contributions. This provides insights into accountability.
The claim is theoretically justified by theorems and proofs in the Appendix. Experiments support the claims; evaluations of the estimation error of the results and the robustness of the noise monotonicity are provided in the Appendix

**Essential References Not Discussed:**

Not that I'm aware of.

**Experimental Designs Or Analyses:**

The experimental designs are sound and clear. The validity of the counterfactual effects have been done on multiple trials. Computational complexity has been discussed. I checked both the Gridworld and the Sepsis experiments.

**Methods And Evaluation Criteria:**

The proposed methods makes sense for the problem of counterfactual effect decomposition in multi-agent systems.
The methods, Shapley values and ICC, provide fair attribution and accountability, and are theoretically sound.
 Evaluation on benchmark datasets (Gridworld and Sepsis) are suitable to the problem.

**Other Comments Or Suggestions:**

N/A

**Other Strengths And Weaknesses:**

The paper addresses significant issues: accountability and explainability of AI systems. It is well written.
One potential weakness would be scalability and the computational complexity involved. To that end, experiments on real-world use cases with many agents would be helpful.

**Questions For Authors:**

The assumption of noise monotonicity guarantees counterfactual identifiability. How realistic is this assumption in real-world use cases and how would the paper's framework treat the cases where the assumption is violated?

**Relation To Broader Scientific Literature:**

The paper draws ideas from various fields, including mediation analysis, intrinsic causal contributions  (Janzing et al), Shapley values, blame attribution (Halpern et al).

**Theoretical Claims:**

I checked the proofs of theorem 3.3. and theorem 4.2. in Appendix H. They look sound to me.

---

> ### Author Rebuttal · Authors · 2025-03-28
>
> Thank you for your valuable feedback and positive score. We are glad to see that you find our paper well-written and addressing significant issues of AI systems. We are also happy to hear that you find our method insightful and sensible, and our experimental setup sound and clear. Please find below our response to your comments and questions.
>
> ## Response to Comments and Questions
>
> **Scalability and Generalizability.** We recognize that improving and evaluating the scalability of our approach would be important for its practical applicability in systems with larger numbers of agents. However, we view this as an independent research challenge that, while important, lies somewhat outside the main scope of this paper. Our primary focus here is to provide an interpretable solution to the problem of counterfactual effect decomposition in multi-agent MDPs.
>
> That said, we note that there are multiple impactful multi-agent settings that naturally involve only a small number of agents, e.g., AI assistants or AI supervision. Nonetheless, empirically evaluating the strategies we propose in Appendix I for mitigating computational complexity, within our setting, represents a promising direction for future work.
>
> **Identifiability Assumptions.** The noise monotonicity assumption imposes a structural restriction on the counterfactual distribution of the causal model. Similar to other causal assumptions, such as *monotonicity* in binary models (Pearl, 1999), noise monotonicity cannot be verified from observational or interventional data. Therefore, assessing its validity in real-world (non-simulated) settings remains challenging.
>
> As you note in your review, we include an empirical robustness analysis in our paper to assess the sensitivity of our findings in the Sepsis experiment to potential violations of noise monotonicity. The results of this analysis are reported in Appendix N.
>
> In Appendix J.1, we discuss extending the applicability of our effect decomposition approach to non-identifiable domains through *partial counterfactual identification* (Manksi, 1990). Rather than relying on point estimates, this approach would operate with bounds on counterfactual quantities. As such, the practical applicability of our method will broaden, albeit on the expense of *efficiency*, i.e., not attributing the full effect.
>
> A partial identification method directly compatible with our setting was recently proposed by (Zhang et al., 2022). Although assessing and enhancing the informativeness of their derived bounds, as well as examining the scalability of their approach in our context, lie beyond the scope of this paper, we regard this as a promising and important avenue for future research.
>
> ## Conclusion
>
> We thank you again for your comments and questions. We would be happy to answer anything else in addition.
>
> ## References
>
> Pearl, Judea. "Probabilities of causation: Three counterfactual interpretations and their identification." Synthese. 1999.
>
> Manski, Charles F. "Nonparametric bounds on treatment effects." The American Economic Review. 1990.
>
> Zhang, Junzhe, Jin Tian, and Elias Bareinboim. "Partial counterfactual identification from observational and experimental data." ICML. 2022.

---

### Official Review · Reviewer_Lft9 · 2025-03-15

**Overall Recommendation:** 4

**Summary:**

This paper focuses on the causal analysis of decision-making in multi-agent cooperative frameworks, specifically the decomposition and attribution of counterfactual effects in the decision-making process of agents. The key contribution is the decomposition of the total counterfactual effect into two parts: the agent-specific effect, which propagates through the subsequent behaviors of other agents, and the state-specific effect, which propagates through environmental state transitions. Furthermore, the authors refine the attribution of the agent-specific effect by leveraging the Shapley value to distribute it among individual agents. The proposed causal analysis method is validated in two experimental environments, demonstrating the feasibility of the counterfactual effect decomposition approach.

### update after rebuttal:
I thank the authors for their response and have taken into account the perspectives of the other reviewers.  I will maintain my score, as it is already a positive one.

**Claims And Evidence:**

The paper's main claim is that its proposed causal analysis method can decompose the counterfactual effect in multi-agent decision processes into effects propagated through agent behaviors and effects propagated through state transitions. The main claim is well-supported by both clear theoretical justifications and empirical results from two experiments, providing sufficient evidence to validate the proposed approach.

**Essential References Not Discussed:**

The paper discusses a sufficient number of related works.

**Experimental Designs Or Analyses:**

I find the experimental setups in the Gridworld and Sepsis environments to be reasonable. Both environments are designed with well-defined agent interactions, and the authors employed posterior sampling and attribution methods to validate the robustness of their proposed framework.

**Methods And Evaluation Criteria:**

I think the proposed methods and evaluation criteria are appropriate. The approach is intuitive, systematically decomposing the total counterfactual effect into agent-specific and state-specific effects, followed by further attribution using Shapley values and ICC. This provides a comprehensive causal analysis framework for multi-agent decision-making.

**Other Comments Or Suggestions:**

N/A

**Other Strengths And Weaknesses:**

One notable innovation in this paper is the introduction of Shapley values to further decompose the total agent-specific effect at the individual agent level. This approach adds granularity to counterfactual effect attribution.

**Questions For Authors:**

N/A

**Relation To Broader Scientific Literature:**

One of the paper’s main contributions is its novel causal analysis perspective on multi-agent cooperation frameworks. Most existing multi-agent cooperation research focuses on designing coordination mechanisms to improve performance on specific tasks. In contrast, this paper provides an analytical tool for attributing decision-making outcomes, which could further enhance the design of cooperative frameworks by enabling a more detailed understanding of agent interactions.

**Theoretical Claims:**

I reviewed the theoretical proofs regarding the decomposition of counterfactual effects, and overall, the proofs appear complete and sound.

---

> ### Author Rebuttal · Authors · 2025-03-28
>
> Thank you very much for your positive score and your kind words about our work. We are glad to hear that you find our approach well-supported, and appreciate its novelty, intuitiveness, and the comprehensive causal analysis it provides for multi-agent decision-making. We are also happy to see that you find our experimental evaluation reasonable and providing sufficient evidence to validate our approach.
>
> Thank you again for all your positive comments. We are happy to answer any further questions or comments you might have.

---

### Official Review · Reviewer_rRtY · 2025-03-24

**Overall Recommendation:** 3

**Summary:**

This paper proposes a novel decomposition of counterfactual effects in multi-agent sequential decision-making settings. Building upon prior work on agent-specific counterfactual effects (cf-ASE), the authors present a bi-level decomposition separating the impact of an agent’s action into (i) how it affects the reachability of outcome-relevant states (r-SSE), and (ii) how much it contributes once those states are reached (ICC).

## Update after rebuttal:
I have read the authors' rebuttal. As I initially gave a weak acceptance, I have decided to maintain my original score.

**Claims And Evidence:**

The authors present a bi-level decomposition of counterfactual effects into reachability (r-SSE) and individual contribution (ICC) terms and formalize this using Structural Causal Models in MMDP settings.
Clear theoretical definitions and mathematical formulation have been shown, and empirical evaluation in environments previously used in causal multi-agent studies (e.g., Sepsis), shows that the decomposition captures interpretable trends across different trust parameters.
However, one caveat is that some claims about practical applicability (e.g., potential use in real-world accountability systems) remain speculative since the experiments are limited to simulated domains, and assumptions such as noise monotonicity may not always hold in practice.

**Essential References Not Discussed:**

None

**Experimental Designs Or Analyses:**

The analysis includes variation across noise models, averaging over rollouts, and decomposition term breakdowns. The error bars and trend lines are presented clearly. In Figure 4, the authors demonstrate how ICC and reachability vary independently, which supports their claim that both aspects are necessary for full interpretation.

However, the environments are simulated and relatively low-dimensional. While the designs are sound for demonstrating the proposed decomposition, it remains unclear how well the method would generalize to high-dimensional and/or real-world domains.

**Methods And Evaluation Criteria:**

The method builds on Structural Causal Models over MMDPs, and introduces a bi-level decomposition into r-SSE (state reachability) and ICC (individual causal contribution). This method is conceptually grounded in prior causal inference frameworks and is logically appropriate for the multi-agent sequential setting.

For evaluation, the authors use two established benchmark environments — Graph and Sepsis — which are well-suited for modeling agent interdependencies and measuring counterfactual effects. Especially, the Sepsis environment (AI-clinician trust model) is directly relevant to applications in human-AI collaboration and accountability.

The limitation is that the experimental evaluation is focused only on simulated environments, and does not include other complex multi-agent RL benchmarks.

**Other Comments Or Suggestions:**

Some key notations appear before being fully explained. Consider adding a notation table or briefly introducing them earlier for clarity.

**Other Strengths And Weaknesses:**

+ Strengths

> The paper builds upon prior work (cf-ASE) and introduces a bi-level decomposition that adds interpretability and granularity to multi-agent causal analysis. While incremental, the conceptual refinement is practically meaningful.

+ Weaknesses

> The evaluation is restricted to relatively simple domains (e.g., Sepsis, Graph), and the decomposition is not tested in complex multi-agent RL benchmarks or real-world tasks. This limits the generalizability of the conclusions.

>  Although well-executed, the core idea is a decomposition of a quantity (cf-ASE) already introduced by the same authors. It seems to be regarded as an incremental extension.

> The paper may be difficult to fully appreciate without reading their ICML 2024 paper, as many definitions and motivations are tightly linked to cf-ASE. This slightly affects self-containment.

**Questions For Authors:**

While the appendix discusses the computational complexity with the increasing number of agents N, are there other challenges (beyond computation) that may arise when scaling the decomposition to large multi-agent systems?

Your proposed decomposition allows for quantitatively separating agent or state influence into r-SSE and ICC. Could the authors provide evidence or examples showing that this separation leads to different decisions or policy adjustments, compared to using cf-ASE alone?

The method assumes access to an accurate SCM or simulator, but real-world models may be misspecified. How robust are r-SSE and ICC to such inaccuracies in structural functions?

While it is common to treat experimental control variables (like the trust level in Sepsis) as fixed scalars, some recent work considers them as a dynamic or latent quantity that evolves over time. Do the authors see the potential for extending their decomposition framework to handle such dynamic control variables, possibly by integrating them into the SCM or policy structure?

**Relation To Broader Scientific Literature:**

This work extends Agent-Specific Effects (ASE) and counterfactual ASE (cf-ASE), introduced in the authors’ prior work (ICML 2024). The current paper decomposes the previously aggregated cf-ASE into two interpretable components,  r-SSE (state reachability) and ICC (individual causal contribution), enhancing the interpretability of causal influence.

This paper leverages SCM formalism applied to sequential settings, building on prior causal RL literature, which uses SCMs to define interventional queries in dynamic systems. However, previous works typically focus on single-agent or non-decomposed effects, whereas this work targets multi-agent interactions with decomposition.

The decomposition proposed in this paper contributes to the growing interest in interpretable multi-agent RL and causal accountability, allowing system designers to trace which agents are causally responsible for which parts of an outcome, a key issue in safety-critical or social contexts.

**Theoretical Claims:**

The introduced theoretical claims appear valid and aligned with established causal inference literature.

---

> ### Author Rebuttal · Authors · 2025-03-28
>
> Thank you for your valuable feedback. We are glad to see that you find our proposed method clear, logically appropriate and enhancing the interpretability of decision-making outcomes. We are also happy to see that you find our experimental testbed well-suited. Please find below our response to your comments and questions.
>
> ## Response to Comments and Questions
>
> **Scalability and Generalizability.** Since Reviewer LcKu had a similar comment, we would like to kindly point you to our response to their review.
>
> **Comparison to ASE.** Our approach to decomposing tot-ASE, termed ASE-SV, builds on the notion of agent-specific effects (ASE) introduced by (Triantafyllou et al., 2024). Fortunately, ASE is already well-defined in the MMDP setting, allowing us to leverage this concept directly. However, it is crucial to clarify that simply summing the agent-specific effects of individual agents does not yield tot-ASE. In fact, our experiments in the Sepsis environment reveal discrepancies of up to 95% in certain scenarios. To derive an *efficient* attribution method, we formulated the problem as a *cooperative game*. This formulation enabled us to define Shapley value in the setting of agent-specific effects. Furthermore, in Section 5 and Appendix F, we explain and formally define in this context a set of well-known desirable properties uniquely satisfied by Shapley value.
>
> Crucially, our work addresses a newly posed and complex problem that is clearly distinct, though related, from the focus of (Triantafyllou et al., 2024). Our key contribution lies in integrating concepts from a wide range of fields, including multi-agent systems (MMDPs), counterfactual reasoning (ASE), mediation analysis (Theorem 3.3), game theory (ASE-SV), information theory (r-SSE-ICC) and more, to tackle this novel challenge.
>
> **Notation Table.** Thank you for your suggestion. We would like to kindly point out that there already exists a notation table in Appendix B of our paper.
>
> **Q1.** While we have not explicitly tested how increasing the number of agents affects the practical performance of our approach, we speculate that scaling to larger multi-agent systems may require drawing more samples from the posterior distribution to ensure reliable counterfactual inference. This is because a larger number of agents increases the dimensionality of the joint action space, which in turn could introduce greater variability in our counterfactual estimates.
>
> **Q2.** Appendix J.2 of our paper demonstrates how the proposed decomposition can be applied to accountable decision making, specifically in the context of *blame attribution* in multi-agent systems. Revisiting the Sepsis scenario from Section 1, we illustrate how our method can be used to determine the proportion of blame attributable to the clinician for their action at time-step 10. Note that this level of fine-grained accountability assessment cannot be achieved by solely examining the total effect and ASE.
>
> **Q3.** First, we would like to clarify that in our Sepsis experiment, we do **not** assume access to the underlying causal model. Instead, our method relies solely on observational distributions. To enable counterfactual inference in this setting, we assume that noise monotonicity holds w.r.t. a chosen set of total orderings. To assess the robustness of our findings in the presence of potential violations of this assumption, we repeated the experiment from Section 6.2 across 5 randomly selected additional total orderings. The full results of this analysis are reported in Appendix N.
>
> These results suggest that while the accuracy of individual counterfactual estimates can degrade when our causal assumptions are violated, the overall conclusions drawn from our analysis in Section 6.2 remain largely robust. This is encouraging, as it provides evidence supporting that our causal explanation formula (including r-SSE and ICC) can yield valuable insights in simulated domains where: (a) effect sizes and randomness reflect real-world settings, and (b) our theoretical assumptions might not hold.
>
> **Q4.** This is an intriguing question, and one we have actually also considered ourselves. The short answer is yes: our effect decomposition framework can be extended to handle dynamic control variables by integrating them into the SCM. One way to do this is by modeling the underlying MMDP as a *mechanised SCM* (Kenton et al., 2023). In this formulation, dynamic control variables can be represented as *mechanism variables* controlling the functional behavior of the MMDP variables. For example, in the Sepsis environment, the clinician's trust level could be modeled not only as evolving over time, but also as being influenced by changes in the AI agent's policy.
>
> Kenton, Zachary, et al. "Discovering agents." Artificial Intelligence. 2023.
>
> ## Conclusion
>
> Thank you again for your comments and questions. We would be happy to answer anything else in addition.

---

### Decision · Program_Chairs · 2025-05-01

**Decision:**

Accept (poster)

**Comment:**

The paper presents a causal explanation framework that decomposes the total counterfactual effect of an agent’s action in multi-agent MDPs. Among the four reviewers, three provided positive evaluations: 2 Accept and 1 Weak Accept. One reviewer raised concerns regarding Theorem, which were addressed step-by-step during the rebuttal; another reviewer explicitly commented that the proof did not appear incorrect.
The authors responded to reviewer concerns throughout the rebuttal phase, though some issues such as the limited experimental scope and the scalability of the approach in real-world settings remain only partially addressed. Taking these factors into account, I would recommend Weak Accept.